# Understanding Multimodal Learning:
# A Loss Landscape Smoothness Perspective

**Jae-Jun Lee**[1]   **Sung Whan Yoon**[1,2]

## Abstract

A surge of recent advancements has consistently highlighted the superiority of multimodal learning over unimodal approaches across a variety of tasks. However, the theoretical foundations elucidating this advantage remain underexplored: existing theoretical analyses are often constrained by tight assumptions, and lack empirical validation. In this paper, we link this gap by proposing a novel theoretical framework grounded in *convolutional smoothing*, offering a new perspective on how multimodal learning contributes to a smoother loss landscape compared to unimodal learning. Building upon this theoretical foundation, we introduce a simple yet effective distributional training approach based on stochastic modality pairing instead of fixed pairing; thus, further promoting flatter landscape via convolutional smoothing. Our empirical results across various multimodal datasets demonstrate that multimodal models not only achieve better performance but also exhibit smoother loss landscape, which represent better robustness and generalization.

## 1. Introduction

Multimodal learning has emerged as a crucial deep learning approach, driven by the increasing demand to integrate diverse modalities of data sources. At its core, multimodal learning aims to combine complementary representations of multiple modalities, such as vision, language, and audio, to achieve richer and better representations. Early approaches primarily focus on fusing modality-specific features either through simple concatenation or via projecting them into a shared latent space. Building upon these foundations, advanced methods have shown meaningful improvements over unimodal baselines by independently regulating the learning rates for each modality (Fujimori et al., 2020) or modulating gradients between modalities (Peng et al., 2022). Concurrently, the recent establishment of large-scale multimodal datasets (Schuhmann et al., 2022) has been accelerating the progress of large-scale models (Vaswani et al., 2017), most notably in the CLIP-based models (Radford et al., 2021). These advancements continue playing a crucial role in pioneering the era of foundation models (Girdhar et al., 2023).

Alongside these empirical advances, efforts to theoretically understand advantages of multimodal learning have been made. A prior theory argues that leveraging multiple modalities can improve training efficiency and generalization by providing a better coverage of the latent feature space compared to unimodal learning (Huang et al., 2021). Alternatively, a recent work offers a mathematical foundation, proving that successful multimodal representation learning is possible when the input data exhibits sufficient heterogeneity and consistency across modalities (Lu, 2023). In addition, one of the very recent works has theoretically proposed that one modality model can benefit synergistically from other modality models by aligning with feature distributions, even without access to the exactly paired annotations across modalities (Lee & Yoon, 2025).

However, these prior theories have clear limitations in providing a fundamental approach, with an apparent coincidence between theory and experiments. They are based on scarce empirical verification on synthetic data and simple datasets, or strong theoretical assumptions for the existence of an ideal representation across modalities. Moreover, prior approaches often handle the generalization bounds or Rademacher complexity analyses; they frequently rely on strong and sometimes restrictive assumptions (Huang et al., 2021; Lu, 2023), which may not hold in practical deep learning settings. Thus, some empirical studies even raise a doubt that multimodality sometimes hinders performance due to challenges such as information overload, leading to overfitting or modality competition (Wang et al., 2020; Huang et al., 2022). To address this limitation, we advocate for a more general theoretical perspective.

[1]Graduate School of Artificial Intelligence, Ulsan National Institute of Science and Technology (UNIST), Ulsan, South Korea. [2]Department of Electrical Engineering, UNIST, Ulsan, South Korea. Correspondence to: Sung Whan Yoon <shyoon8@unist.ac.kr>.

*Proceedings of the 43rd International Conference on Machine Learning*, Seoul, South Korea. PMLR 306, 2026. Copyright 2026 by the author(s).

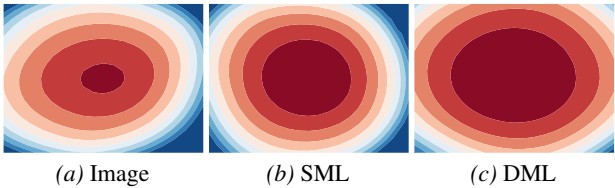

*(a)* Image      *(b)* SML      *(c)* DML

*Figure 1.* **Loss landscape visualizations for AVMNIST dataset.** This figure presents a 2D projection of the loss landscape, following the loss landscape visualization proposed in (Li et al., 2018).

Hence, we present a novel perspective beyond these conventional theoretical viewpoints by interpreting multimodal learning through the lens of loss landscape flatness. Specifically, we propose that the presence of additional modalities induces a form of *convolutional smoothing* over the loss landscape associated with unimodal learning. Intuitively, for data $\mathbf{x}_i$ from modality $i$ paired with $\mathbf{x}_j$ from modality $j$, multimodal learning naturally smooths the loss surface by minimizing the expected loss under the joint distribution $p(\mathbf{x}_i, \mathbf{x}_j)$, which is analogous to a convolution. This smoothing effect reduces landscape sharpness, effectively guiding optimization toward flatter minima, which are often associated with improved robustness. Notably, this behavior arises intrinsically from multimodal training itself, without explicitly applying techniques designed to promote flatness, highlighting it as a novel perspective. Building on this insight, we further introduce a simple training approach, **D**istributional **M**ultimodal **L**earning (**DML**), which leverages this smoothing effect by stochastically pairing samples between modalities $i$ and $j$ within a shared output space $\mathcal{Y}$.

To offer an intuitive teaser, we visualize the loss landscape in a preliminary experiment. As shown in Figure 1, multimodal learning on the AVMNIST dataset with image (vision) and audio modalities yields significantly smoother and flatter loss landscapes than unimodal learning. More importantly, the visualization highlights that DML consistently converges to flatter minima than standard multimodal learning, where inputs are strictly paired one-to-one. This suggests that stochastic pairing in DML further strengthens the convolutional smoothing effect, biasing optimization toward flatter regions of the landscape. Taken together, these results underscore the potential of our hypothesis to improve robustness, particularly under diverse modality pairings.

## 2. Related Work

**Fusion Approaches in Multimodal Learning** Multimodal learning aims to enhance model performance by integrating information from different modalities, such as vision, audio, and text. Among the various fusion strategies, early fusion has gained significant attention in recent works (Baltrusaitis et al., 2019). On late fusion, each modality is processed independently through dedicated encoders and fused at a late stage, which preserves modality-specific representations

and supports flexible alignment at the decision level (Zadeh et al., 2017; Atrey et al., 2010). Early fusion focuses on combining input-level or representation-level information from multiple modalities, allowing joint processing of the model's features, which can potentially capture deeper interactions between modalities (Ngiam et al., 2011; Barnum et al., 2020). Recent methods enhance this strategy by incorporating cross-modal attention and transformer-based architectures to improve interactions across modalities (Tsai et al., 2019; Nagrani et al., 2021). In this paper, we adopt an early fusion with concatenation, which enables a clearer analysis of the representation level from each modality.

**Flat Minima and Robustness in Deep Learning** Finding flatter minima has emerged as a main key concept for understanding robustness and generalization in deep learning. For instance, prior methods such as Stochastic Weight Averaging (SWA) (Izmailov et al., 2018) and its dense variant (SWAD) (Cha et al., 2021) employ heuristic approach for locating flatter regions of the loss landscape by averaging model parameter weights. More recent approaches, such as Sharpness-Aware Minimization (SAM) and its extensions (Foret et al., 2021; Kwon et al., 2021), minimize the worst-case loss within a neighborhood of the current parameters and thereby converge to flatter loss landscape.

However, despite the importance of flatness, the role of loss landscape in multimodal learning has received relatively limited attention, especially from a theoretical perspective, or uses just to highlight their empirical robustness gains (Wei & Hu, 2024). Therefore, we aim to deepen this understanding by analyzing the behavior of multimodal learning from the perspective of loss landscape. Through both qualitative and quantitative results, we show that multimodal learning essentially leads to flatter minima.

**Comparison with Data Augmentation (DA)** Recent study has suggested that standard DA, such as jittering, flipping or adding gaussian noise, naturally smooth the loss landscape by perturbing the input space (Bishop, 1995; Yoo & Yoon, 2025). From the perspective of manipulating input data, this seems similar to stochastically pairing modality data. However, we clarify that our novel approach is fundamentally different. Specifically, DA is *limited to local perturbations* within a single modality. In contrast, our approach pairs data across different modalities and *explores output space*.

## 3. Preliminaries and Problem Setup

### 3.1. Notation

Before proposing main theories, we first introduce basic notations to be used in our theoretical analyses.

- $\mathbf{x}_i \in \mathcal{X}_i \subset \mathbb{R}^{d_i}, \mathbf{x}_j \in \mathcal{X}_j \subset \mathbb{R}^{d_j}$ denote input data from two different modalities $i$ and $j$, with their data distribution $\mathbf{x}_i \sim p(\mathbf{x}_i)$ and $\mathbf{x}_j \sim p(\mathbf{x}_j)$, respectively.

- $y \in \mathcal{Y}$ be the target output, which indicates the shared semantics / output space across modalities.

- $f_i(\mathbf{x}_i), f_j(\mathbf{x}_j) \in \mathbb{R}^d$ are the modality encoders for the respective modality.

- $f(\mathbf{x}_i, \mathbf{x}_j; \theta) := \phi(f_i(\mathbf{x}_i), f_j(\mathbf{x}_j))$ be a function parametrized by $\theta \in \mathbb{R}^m$, with modality specific function $f_i(\cdot)$ and $f_j(\cdot)$, and modality fusion function $\phi(\cdot, \cdot)$.

- $\ell(f(\cdot; \theta), y)$ is the loss function.

### 3.2. Local Linearization of $\phi$: The ATI Property

Given the critical role of the modality fusion function $\phi$ in multimodal learning, we characterize the behavior of the $\phi$. Specifically, we posit that $\phi$ admits approximate linearization, implying that it satisfies the Approximate Translation-Invariance (ATI) property.

Let $\phi : \mathbb{R}^{d_i} \times \mathbb{R}^{d_j} \to \mathbb{R}^d$ be a mapping function that fuses representation vectors from two modalities. Consider a small perturbation $\boldsymbol{\tau}$ applied to the second modality $\mathbf{v}$. Then, for a sufficiently small $\boldsymbol{\tau}$, the output of $\phi$ can be locally approximated as:

$$\phi(\mathbf{u}, \mathbf{v} + \boldsymbol{\tau}) \approx \phi(\mathbf{u}, \mathbf{v}) + \boldsymbol{\alpha} \odot \boldsymbol{\tau}, \qquad (1)$$

where $\boldsymbol{\alpha} \in \mathbb{R}^d$ represents the local sensitivity of $\phi$ with respect to $\mathbf{v}$. This demonstrates that the fusion function behaves approximately linearly within a local neighborhood. We define this as the ATI property, where output shift is linear to the input perturbation. Here, $\boldsymbol{\alpha}$ usually corresponds to the (diagonal of the) Jacobian $J_\phi$, reflecting a first-order Taylor approximation even in non-linear networks.

To empirically validate this local linearization (and thus the ATI property) in standard fusion architectures, we conducted a toy experiments. We added random perturbations $\boldsymbol{\tau}$ to $\mathbf{v}$ across input pairs $(\mathbf{u}, \mathbf{v})$ and evaluated four common fusion mechanisms: additive fusion (**Add.**), a linear MLP (**Lin.**), a non-linear MLP with ReLU (**Non-Lin.**), and an 8-head cross-attention block (**Cross-Atten.**). We then quantify the deviation from linearity using the invariance error $\Delta_{\text{error}} = \|\phi(\mathbf{u}, \mathbf{v} + \boldsymbol{\tau}) - \phi(\mathbf{u}, \mathbf{v}) - \alpha \odot \boldsymbol{\tau}\|_2$ and the relative error $\epsilon_{\text{rel}} = \Delta_{\text{error}} / \|\phi(\mathbf{u}, \mathbf{v})\|_2$.

*Table 1.* **Testing Linearity for Various Fusion Networks**

| Metric | Add. | Lin. | Non-Lin. | Cross-Atten. |
|---|---|---|---|---|
| Feat. Norm | 1.4103 | 0.6220 | 0.6839 | 0.4267 |
| $\Delta_{\text{error}}$ | 0 | 0.0047 | 0.0067 | 0.0051 |
| $\epsilon_{\text{rel}} \times 100$ | 0 | 0.76 % | 0.98 % | 1.20 % |

The results in Table 1 demonstrate that all fusion strategies exhibit minimal deviation from the linear approximation. As expected, additive fusion satisfies the ATI property exactly. Crucially, both MLP-based variants and Cross-Attention

exhibit small errors ($\epsilon_{\text{rel}} < 1.2\%$), confirming that the ATI property holds approximately even in complex non-linear networks. These findings empirically support the validity of the linearization on Equation (1), thereby confirming that the ATI property is also satisfied. Empirical justification of ATI in deeper layers and mathematical derivations of the first-order Taylor expansion are provided in Appendix A.3.

### 3.3. Multimodal Learning Objective Function

In this section, we formulate the multimodal learning setup. As previously denoted on Section 3.1, let $\mathbf{x}_i$ and $\mathbf{x}_j$ denote input data sampled from the distributions of modalities $i$ and $j$, respectively, and let $y \in \mathcal{Y}$ represent shared semantics (*e.g.*, same label, shared representation space).

Conventionally, multimodal learning relies on strictly paired data, where each instance $(\mathbf{x}_i, \mathbf{x}_j)$ is drawn from a joint distribution observing the same pairs. However, we propose a more generalized setting based on the conditional independence settings. Specifically, Given the shared semantic $y$, the joint distribution of the modalities can be factorized as:

$$p(\mathbf{x}_i, \mathbf{x}_j \mid y) = p(\mathbf{x}_i \mid y) p(\mathbf{x}_j \mid y), \qquad (2)$$

which corresponds to sampling each modality independently given the output $y$. Equation (2) implies that as long as the modalities share the same target semantic $y$, they can be sampled independently from their respective conditional distributions, rather than being restricted to fixed, pre-aligned pairs. Crucially, this setting does not deviate from the fundamental goal of multimodal learning; it merely relaxes the data pairing constraint while preserving the semantic alignment required between modalities. We provide detail discussion in Section 7. Based on this formulation, we define the expected uni-and mult-modal losses as follows:

$$\mathcal{L}_{\text{uni}} = \mathbb{E}_{y \in \mathcal{Y}} \left[ \int \ell(f(\mathbf{x}_i, \bar{\mathbf{x}}_j; \theta), y) p(\mathbf{x}_i \mid y) \, d\mathbf{x}_i \right], \qquad (3)$$

$$\mathcal{L}_{\text{multi}} = \mathbb{E}_{y \in \mathcal{Y}} \left[ \iint \ell(f(\mathbf{x}_i, \mathbf{x}_j; \theta), y) p(\mathbf{x}_i, \mathbf{x}_j \mid y) \, d\mathbf{x}_i \, d\mathbf{x}_j \right], \qquad (4)$$

where $\bar{\mathbf{x}}_j$ represents the absence of modality $j$ (*e.g.*, a zero vector). For the convenience, we denote expected losses as $\mathcal{L}_{\text{uni}} = \mathbb{E}_y[\mathcal{L}_{\text{uni}}(f \mid y)]$ and $\mathcal{L}_{\text{multi}} = \mathbb{E}_y[\mathcal{L}_{\text{multi}}(f \mid y)]$.

## 4. Theoretical Analysis: Convolutional Smoothing in Multimodal Learning

In this section, we provide the main theories of the *convolutional smoothing* in multimodal learning, which naturally appears to reach to flatter minima than unimodal learning.

## 4.1. Main Theory: Convolutional Smoothing Effect

In this section, we introduce that the multimodal loss can be interpreted as a convolutional smoothing of the unimodal loss under the property of ATI of $\phi$ (*i.e.*, Equation (1)) and the condition of paired modalities (*i.e.*, Equation (2)). Before deriving the convolutional form of the multimodal loss, we first define the concepts of the fusion-induced scaled shift and its corresponding kernel.

**Definition 1** (**Fusion-Induced Scaled Shift and Kernel**). *Let $\bar{\mathbf{x}}_j$ denote input data representing absence of modality $j$, such zero vector or constant value, and define fusion-induced scaled shift as with coefficient $\boldsymbol{\alpha} = \alpha(\mathbf{x}_i, \mathbf{x}_j)$:*

$$\tau_{\boldsymbol{\alpha}}(\mathbf{x}_j) := \boldsymbol{\alpha} \odot (\phi(f_i(\mathbf{x}_i), f_j(\mathbf{x}_j)) - \phi(f_i(\mathbf{x}_i), f_j(\bar{\mathbf{x}}_j))),$$

*(5)*

*The corresponding* scaled-shift kernel *is then defined using the Dirac delta function $\delta_{\tau_{\boldsymbol{\alpha}}(\mathbf{x}_j)}(\cdot)$ centered at $\tau_{\boldsymbol{\alpha}}(\mathbf{x}_j)$:*

$$\mathcal{K}_{\mathbf{x}_j, \boldsymbol{\alpha}}(\boldsymbol{\tau}) := \mathop{\mathbb{E}}_{\mathbf{x}_j \sim p(\mathbf{x}_j | y)} \left[ \delta_{\tau_{\boldsymbol{\alpha}}(\mathbf{x}_j)}(\boldsymbol{\tau}) \right], \quad (6)$$

We present the following theorems, which establish that the multimodal learning loss can be expressed as a convolution of the unimodal loss with the scaled-shift kernel, $\mathcal{K}_{\mathbf{x}_j, \boldsymbol{\alpha}}$. For the justification of Definition 1, see Appendix C.1.

**Theorem 1** (**Convolutional Smoothing with Modality Scaled Shifted Kernel**). *Under the Equation (1), the expected multimodal loss is a scaled convolution of the unimodal loss:*

$$\mathcal{L}_{multi} = \mathop{\mathbb{E}}_{y \in \mathcal{Y}} \left[ \left( \mathcal{L}_{uni} \circledast \mathcal{K}_{\mathbf{x}_j, \boldsymbol{\alpha}} \right) (f \mid y) \right]. \quad (7)$$

**Theorem 2** (**Loss Landscape of Multimodal Learning**). *Suppose that conditional unimodal loss $\mathcal{L}_{uni}(f \mid y)$ is continuous and bounded for all $y \in \mathcal{Y}$, and the shift kernel $\mathcal{K}_{\mathbf{x}_j, \boldsymbol{\alpha}}$ has finite variance. Then the $\mathcal{L}_{multi}$ is smoother than the $\mathcal{L}_{uni}$, in the following two folds:*

1. ***Upper Bound on the Spectral Norm of the Hessian:*** *For any $f$, the spectral norm of the multimodal loss Hessian is upper-bounded by that of the unimodal loss Hessian:*

$$\sup_f \|\nabla^2 \mathcal{L}_{multi}(f \mid y)\|_2 \leq \sup_f \|\nabla^2 \mathcal{L}_{uni}(f \mid y)\|_2. \quad (8)$$

2. ***Frequency Domain Interpretation:*** *Let $\widehat{\mathcal{L}}_{multi}(\omega)$, $\widehat{\mathcal{L}}_{uni}(\omega)$, and $\widehat{\mathcal{K}}_{\mathbf{x}_j, \boldsymbol{\alpha}}(\omega)$ denote the Fourier transforms of $\mathcal{L}_{multi}$, $\mathcal{L}_{uni}$, and $\mathcal{K}_{\mathbf{x}_j, \boldsymbol{\alpha}}$, respectively. Then, Equation (7) can be expressed in frequency domain as: $\widehat{\mathcal{L}}_{multi}(\omega) = \widehat{\mathcal{L}}_{uni}(\omega) \cdot \widehat{\mathcal{K}}_{\mathbf{x}_j, \boldsymbol{\alpha}}(\omega)$. Moreover, it follows that:*

$$\mathcal{O}(\widehat{\mathcal{L}}_{multi}(\omega)) \leq \mathcal{O}(\widehat{\mathcal{L}}_{uni}(\omega)). \quad (9)$$

The proofs of Theorems 1 and 2 are provided in Appendix A. We now present Remark 2.1 and Remark 2.2, offering theoretical insights and their intuitions into the fundamental properties of multimodal learning how they leads to smoother loss landscape:

**Remark 2.1** (**Worst-Case Smoothness Comparison: Equation** (8)). *The **hessian spectral norm** is widely used to quantify the smoothness of the loss landscape (Ghorbani et al., 2019), where lower eigenvalues indicates flatter region. In this context, our analysis establishes a key theoretical guarantee: the worst-case Hessian spectral norm of a multimodal loss, $\mathcal{L}_{multi}$ is upper-bounded by that of a unimodal loss, $\mathcal{L}_{uni}$. This view shows that **multimodal learning approach at least have smoother or equal loss landscape under worst-case conditions** compare to unimodal.*

**Remark 2.2** (**Multimodal as a Low-Pass Filter in the Frequency Domain: Equation** (9)). *The smoothing mechanism of multimodal learning can be elucidated from a **frequency-domain** perspective: $\mathcal{L}_{multi}$ attains smoother minima due to the influence of the scaled-shift kernel $\widehat{\mathcal{K}}_{\mathbf{x}_j, \boldsymbol{\alpha}}$. In particular, when the underlying distribution of $\mathcal{K}_{\mathbf{x}_j, \boldsymbol{\alpha}}$ exhibits sufficient variance, its Fourier transform $\widehat{\mathcal{K}}_{\mathbf{x}_j, \boldsymbol{\alpha}}$ decays rapidly as the frequency magnitude $\|\omega\| \to \infty$. This decay acts as a low-pass filter that **effectively suppresses the high-frequency components** of $\widehat{\mathcal{L}}_{uni}(\omega)$. Consequently, $\mathcal{L}_{multi}$ generally exhibits a smoother and flatter loss landscape compared to $\mathcal{L}_{uni}$ in frequency domain perspective.*

To support the validity of our theoretical framework, we emphasize a key consideration: the latent representations $f_i(\mathbf{x}_i)$ and $f_j(\mathbf{x}_j)$ should exhibit meaningful differences. This aligns with prior theoretical work emphasizing the importance of modality heterogeneity, that modalities should be different, but not arbitrarily so—for effective multimodal learning (Lu, 2023). For instance, in extreme cases where one modality consists of random Gaussian noise, its contribution becomes negligible, making it practically irrelevant. To better illustrate these theoretical frameworks, we present a conceptual illustration of our framework in Figure 2.

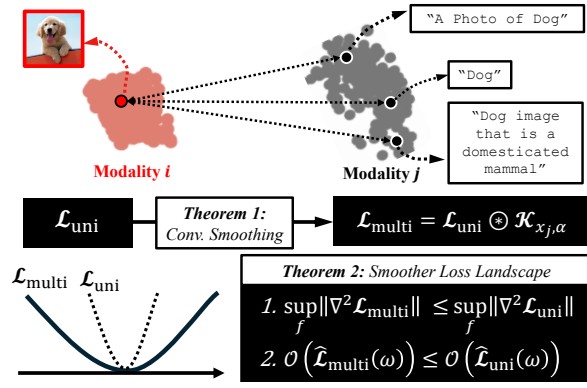

*Figure 2.* A conceptual sketch of our theoretical frameworks.

## 4.2. Distributional Multimodal Learning (DML) Beyond Pointwise Pairing

Standard multimodal learning typically relies on pointwise-paired datasets, emphasizing exact correspondences between modalities under the empirical risk minimization

| Settings | Modality $i$ | Modality $j$ |
|---|---|---|
| **SML** (Pointwise) 

 **DML** (Stochastic) | 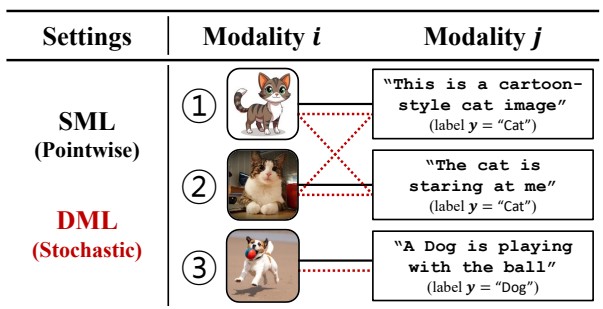 | |

*Figure 3.* Exactly Paired (SML) vs. Stochastically Paired (DML). DML indicates possible multimodal pairings $(\mathbf{x}_i, \mathbf{x}_j)$, which are stochastically matched under given $y$. This indicates that, in DML, the two modalities are sampled independently conditioned on $y$.

framework. From a representation learning perspective, this approach leverages modality-specific information only at the level of individual data points. While it may implicitly capture certain aspects of modality-specific distributions, training remains limited as supervision is constrained to pointwise alignments. In particular, strict pointwise pairing restricts supervision to instance-level alignment in the output space $\mathcal{Y}$, thereby biasing the learned hypothesis $h_\theta : \mathcal{X} \to \mathcal{Y}$ toward solutions that align with $y$ only through paired instances and limiting multimodal representations.

To address this limitation, we propose a simple modification on pairing modality: shuffling datapoints across modalities within the same label space. This encourages stronger distributional alignment and is *directly motivated by our theoretical framework* based on given Equation (2). We refer to this approach as **D**istributional **M**ultimodal **L**earning (**DML**), as illustrated in upper part of Figure 2 and Figure 3. For clarity, we also refer to the conventional pointwise-paired setting as **S**tandard **M**ultimodal **L**earning (**SML**).

As shown in Figure 3, let us consider a classification task involving "dog" and "cat" classes. The figure presents paired instances of modality $i$ (image) and modality $j$ (text) associated with labels $y$. While **SML** is restricted to training on the originally aligned pairs (**black solid lines**), **DML** facilitate a stochastic pairing approach (**red dotted lines**). Specifically, DML randomly matches instances across modalities provided they share the same label $y$, even if their specific descriptions are slightly different. This strategy remains valid because the objective is class-level discrimination; for instance, the samples in rows ① and ② are interchangeable as they both share the semantic label $y = $ "cat".

Intuitively, DML encourages the model to align class-level distributions rather than relying on exact data-level correspondences. Specifically, this approach aligns closely with our theoretical insights. By stochastically pairing data during training, DML implicitly increases the variance of the input distribution, resulting in broader supervision. This aligns with our theoretical insights provided by Theorem 2, where increasing the variance of the data distribution leads

to a smoother loss landscape. In other words, DML could possibly maximize the variance of the kernel $\mathcal{K}_{\mathbf{x}_j, \boldsymbol{\alpha}}$ by sampling $x_j$ from the entire marginal distribution $p(x_j \mid y)$, which conceptually leads to the highest entropy. Then this high-variance kernel is precisely what Remark 2.2 requires to serve as a powerful low-pass filter, effectively smoothing the loss landscape by suppressing high-frequency components from a frequency-domain perspective.

# 5. Experiments

Here, we present experimental results comparing unimodal and multimodal learning through a lens of the loss landscape, along with descriptions of the datasets, settings, and analyses of the results. The detailed descriptions of the datasets, hyperparameter configurations, and other additional experimental settings are provided in Appendix B[1].

### 5.1. Experimental Setup

**Multimodal Datasets** We conducted experiments and analyses on four different multimodal datasets: **Kinetics-Sounds** (Arandjelovic & Zisserman, 2017), **AVMNIST** (Vielzeuf et al., 2018), **CREMA-D** (Cao et al., 2014), and **UPMC-Food101** (Wang et al., 2015). **Kinetics-Sounds** is an Audio-Visual [A+V] dataset partially selected 31 classes from the Kinetics-400 dataset. **AVMNIST** is an Image-Audio [I+A] dataset comprising of MNIST images, paired with 112×112 spectrograms generated from spoken digit audio. **CREMA-D** is an Audio-Visual [A+V] dataset within 6 classes, containing facial and vocal expressions. Lastly, **UPMC-Food101** is an Image-Text [I+T] dataset featuring 101 food categories with recipe texts. The details are provided in Appendix B.1.

**Model Architectures** We follow standard model architectures from prior works (Peng et al., 2022; Wei & Hu, 2024) of multimodal learning, just slightly modifying them to each dataset and modality. For Kinetic-Sound, AVMNIST and CREMA-D, we use a ResNet-based architecture (He et al., 2016), where audio data is transformed into spectrogram images. For UPMC-Food101, we use a pretrained BERT model (Devlin et al., 2019) as the text encoder and a pretrained ImageNet-1k (Krizhevsky et al., 2012) ResNet-18 model as the vision encoder by following (Xu et al., 2025).

**Hyperparameter Settings** We mostly followed the standard hyperparameter settings used in multimodal learning (Xu et al., 2025) and additionally tuned for more careful training. Training is conducted for 70 epochs across all experiments. We apply the SGD optimizer (Bottou, 1998) as done in the previous works, and partially replace to the Adam optimizer (Kingma & Ba, 2015). Additional hyperparameters and detailed descriptions are provided in Appendix B.2.

---

[1]The official code is available here.

*Table 2.* **Classification accuracies on multimodal datasets.** For each dataset, we denote $i$ modality as dominant if it outperforms the other modality. For datasets with three modalities, we denote $i$ modality as the first and $j$ modality as the second most dominant, respectively, based on their relative performance. All results are averaged over 3 trials, with standard deviations reported.

| Datasets | Kinetics-Sounds | | AVMNIST | | CREMA-D | | UPMC-Food101 | |
|---|---|---|---|---|---|---|---|---|
| | Type | Accuracy | Type | Accuracy | Type | Accuracy | Type | Accuracy |
| Modality $i$ | [A] | $51.95 \pm 0.86$ | [I] | $64.98 \pm 0.16$ | [A] | $60.19 \pm 2.19$ | [T] | $86.38 \pm 0.11$ |
| Modality $j$ | [V] | $45.70 \pm 0.77$ | [A] | $41.84 \pm 0.24$ | [V] | $48.71 \pm 1.58$ | [I] | $64.68 \pm 0.28$ |
| SML | [V+A] | $62.95 \pm 0.51$ | [I+A] | $70.30 \pm 0.12$ | [A+V] | $64.95 \pm 1.40$ | [T+I] | $91.48 \pm 0.08$ |
| DML | [V+A] | $\mathbf{65.89 \pm 0.50}$ | [I+A] | $\mathbf{71.69 \pm 0.08}$ | [A+V] | $\mathbf{65.53 \pm 1.31}$ | [T+I] | $\mathbf{92.84 \pm 0.05}$ |

## 5.2. Main Performance Results

In this section, we present results from reproduced experiments comparing unimodal learning, SML, and our empirical suggestion, i.e., DML, across several benchmark datasets. For each dataset, we report classification accuracy for individual modalities, SML, and DML. The dominant modalities are chosen based on their unimodal performance, with $i$ representing the most dominant modality.

Our findings based on Table 2 demonstrate the power of multimodal learning, especially the proposed DML, which stochastically matches samples. Especially, the combination of [A] and [V] in **Kinetics-Sounds** produces a substantial performance increase over single modalities. While [A] achieves 51.95% accuracy individually, the multimodal approaches yield substantial improvements: SML reaches 62.95% (+11.00%), and DML advances to 65.89% (+13.94%). The gain of DML (+2.94% over SML) shows effective exploitation of multimodal distributional alignment.

A similar trend appears in **AVMNIST**, where the dominant modality [I] attains 64.98% accuracy. Fusing [I] and [A] with SML improves performance to 70.30% (+5.32%), and applying DML yields an additional gain to 71.69% (+6.71%). This suggests that distributionally matched supervision benefits rather than exactly matched pair over given modalities. For **CREMA-D**, the benefits are more modest but still consistent. [A] provides the highest unimodal accuracy at 60.19%, and SML elevates this to 64.95% (+4.76%). DML achieves 65.53% (+5.34%), consistent improvement, and continues to surpass individual modalities and SML.

This pattern indicates that multimodal learning consistently outperforms single modalities, and even under moderate modality complementarity, class-level stochastic alignment can extract incremental gains. Finally, in **UPMC-Food101**, [T] alone achieves a strong baseline of 86.38%, but its combination with [I] under SML boosts accuracy to 91.48% (+5.10%), and DML increases up to 92.59% (+6.46%).

These findings demonstrate that SML consistently outperforms the dominant unimodal baselines across all datasets. Furthermore, DML achieves additional gains by aligning modality distributions at the class level, thereby leveraging

*Table 3.* Performance with Data Augmentation

| DA | KS | AVMNIST | CREMA-D | Food101 |
|---|---|---|---|---|
| SML | 62.95 | 70.35 | 64.95 | 91.48 |
| + J + C + F | 64.51 | 71.89 | 65.92 | 91.93 |
| DML | 65.89 | 71.69 | 65.53 | 92.84 |
| + J + C + F | **67.19** | **73.01** | **67.39** | **94.93** |

J: Color Jitter, C: Cropping, F: Flipping

diverse information for each modality model. These results strongly support our theoretical view that multimodal learning improves performance through facilitating flatter optimization landscapes relative to unimodal learning.

## 5.3. Comparison with Explicit Smoothing Methods

Here, we conducted additional experiments to demonstrate how the DML approach can complement additional techniques, such as Data Augmentation (DA), and to highlight its novelty compared to flatness-based approaches.

**Data Augmentation (DA)** DA is a widely used technique in deep learning and is known to have a smoothing effect (Yoo & Yoon, 2025). In this sense, DML may appear similar to DA, since the number of possible pairings increases during training and also have smoothing effect based on Theorem 2. However, as mentioned in Section 2, DML is clearly different from DA, and is even complementary.

To validate this, we conducted experiments demonstrating their complementary effect. As shown in Table 3, DA consistently improves both SML and DML. In particular, applying DA to DML yields additional gains over vanilla DML, improving performance by +1.30% on Kinetics-Sounds, +1.32% on AVMNIST, +1.86% on CREMA-D, and +2.09% on UPMC-Food101. Due to space limitations, we further clarify how DML and DA differ and how they can complement each other, with additional results in Appendix C.3.

**Flatness Methods** We compare our approach to recent flatness-based methods to evaluate how DML maintains novelty in multimodal learning relative to smoothing techniques. Specifically, we applied Label Smoothing (LS) (Szegedy et al., 2016), Stochastic Weight Averaging (SWA) (Izmailov et al., 2018), and Sharpness-Aware Minimization (SAM) (Foret et al., 2021), to SML and compare with DML.

*Table 4.* Comparison of Flatness Approaches and DML

| Methods | KS | AVMNIST | CREMA-D | Food101 |
|---|---|---|---|---|
| SML | 62.95 | 70.30 | 64.95 | 91.48 |
| SML + LS | 63.03 | 70.44 | 64.79 | 91.79 |
| SML + SWA | 64.88 | 69.24 | 64.34 | 92.08 |
| SML + SAM | 64.19 | 70.91 | **65.59** | 91.76 |
| DML | **65.89** | **71.69** | 65.53 | **92.84** |

As shown in Table 4, DML consistently shows better performances compared to SML + flatness approaches across most datasets, with the exception of CREMA-D. Across these three datasets, the improvements range approximately from +1.1% to +1.7%, reflecting consistent gains beyond what those achieved by LS, SWA, or SAM. This demonstrates that DML not only benefits from a smoothing effect but also outperforms compared to recent flatness approaches.

## 5.4. Image-Text Retrieval

DML has been primarily evaluated on supervised classification tasks, but its core principle—leveraging shared semantic or feature-level information conditioned on label-like signals—is not limited to supervised settings. In fact, DML can be extended to self-supervised frameworks, such as contrastive learning (*e.g.*, CLIP (Radford et al., 2021)), where pairing can be guided by pseudo-labels or semantic similarity rather than ground-truth annotations.

To validate this, we applied DML to a contrastive learning setup using the **Flickr-30k** dataset (Plummer et al., 2015). Since Flickr-30k does not provide the explicit class annotations required by DML, we constructed pseudo-labels through the following procedure. We first extracted image features using a pretrained ViT-L/14 model, and then applied UMAP (McInnes et al., 2018) to obtain a low-dimensional space that preserves the intrinsic representational geometry, as correlations and structural relationships are not readily observable in the original high-dimensional space. Then, we applied $K$-means clustering with $K = 1000$ to partition the embedding space. Each image is assigned to the nearest cluster centroid, which we treated as its pseudo-class label.

This pseudo-labeling scheme enabled the construction of a shared output space that captures semantic similarities between data points. We trained the model using the CLIP loss and evaluated performance on both **Image-to-Text (I2T)** and **Text-to-Image (T2I)** retrieval tasks. For the detailed hyperparameter settings, see Appendix B.2.

As the results, Table 5 shows that DML consistently improves most retrieval metrics. For I2T, DML increases R@1 from 87.34% to 89.10% (+1.76%) and R@5 from 97.10% to 97.56% (+0.46%) compared to SML. A similar trend appears in T2I, with R@1 and R@5 rising to 76.10% (+1.91%) and 90.51% (+0.74%), respectively. This demonstrates that stochastic alignment is an effective method for improving

*Table 5.* Retrieval performance of SML and DML on Flickr–30k. Metrics are reported as Recall@K (%).

| Tasks | Image-to-Text | | Text-to-Image | |
|---|---|---|---|---|
| **Metrics** | R@1 | R@5 | R@1 | R@5 |
| SML | 87.34 | 97.10 | 74.19 | 89.77 |
| **DML** | **89.10** | **97.56** | **76.10** | **90.51** |

retrieval results, even without explicit class supervision.

## 6. Further Analysis

### 6.1. Loss Landscape Flatness and Smoothness

Here, we employ two widely recognized metrics of flatness to evaluate generalization and robustness: the *Hessian maximum eigenvalue*, which reflects the local curvature of the loss surface, and the *Low-Pass Filter (LPF) metric* that quantifies the stability of the trained model under perturbations.

**Hessian Spectrum Analysis** The *Hessian of maximum eigenvalue* (Ghorbani et al., 2019; Yao et al., 2020) quantifies the sharpest curvature direction of the loss landscape, with smaller values indicating flatter minima. Yet, unimodal and multimodal models differ in architecture and parameter count; direct comparison of absolute Hessian eigenvalues could be misleading. To facilitate a more comparable assessment, we additionally report the ratio between the largest and 5th eigenvalues ($\lambda_{\max}/\lambda_5$), as a common surrogate for the smoothness of loss curvature (Jastrzebski et al., 2020; Foret et al., 2021). Specifically, this ratio captures the anisotropy of the curvature and quantifies the local shape of the loss landscape; lower values indicate a flatter.

**LPF Metric** The *LPF metric* (Bisla et al., 2022) complements this analysis by quantifying the sensitivity of the loss surface to perturbations via Gaussian smoothing. Specifically, we define $\Delta_{\text{LPF}}$ as the difference between the original loss $\mathcal{L}$ and the smoothed loss $\mathcal{L}_{\text{LPF}}$, obtained by convolving the loss landscape with a Gaussian filter. This discrepancy reflects the susceptibility of the loss landscape to perturbations; a smaller $\Delta_{\text{LPF}}$ indicates greater robustness.

**Flatness Results** We compare these generalization and robustness metrics across three settings: the dominant unimodal modality, SML, and DML. As expected, the results in Table 6 largely align with our theoretical hypothesis—multimodal approaches generally lead to smoother loss landscapes and improved robustness. Prominently, DML consistently outperforms other methods across most multimodal datasets. For example, results on Kinetics-Sounds, AVMNIST, and UPMC-Food101 datasets outperformed all baselines, where multimodal approaches obviously show lower flatness values compared to unimodal. Moreover, DML also usually exhibits lower values on given metrics compared to SML, where training with stochastically

*Table 6.* **Flatness and robustness across various multimodal datasets.** Here, lower values (↓) indicate better performance.

| KS | $\lambda_{\max}$ (↓) | $\lambda^*$ (↓) | $\mathcal{L}_{\mathrm{LPF}}$ (↓) | $\Delta_{\mathrm{LPF}}$ (↓) | AVMNIST | $\lambda_{\max}$ (↓) | $\lambda^*$ (↓) | $\mathcal{L}_{\mathrm{LPF}}$ (↓) | $\Delta_{\mathrm{LPF}}$ (↓) |
|---|---|---|---|---|---|---|---|---|---|
| Modality $i$: [A] | 13.8997 | 1.4284 | 2.1848 | 0.4781 | Modality $i$: [I] | 7.9102 | 2.9235 | 0.9829 | 0.0138 |
| Modality $j$: [V] | 10.4048 | 1.5235 | 2.8545 | 0.2083 | Modality $j$: [A] | 9.3102 | 1.5225 | 1.9924 | 0.0021 |
| SML | 9.9691 | 1.4287 | 1.6703 | 0.3097 | SML | 6.2560 | 1.4338 | 0.9129 | 0.0121 |
| DML | **9.5732** | **1.1686** | **1.4637** | **0.1659** | DML | **5.6220** | **1.2492** | **0.8083** | **0.0083** |
| **CREMA-D** | $\lambda_{\max}$ (↓) | $\lambda^*$ (↓) | $\mathcal{L}_{\mathrm{LPF}}$ (↓) | $\Delta_{\mathrm{LPF}}$ (↓) | **Food101** | $\lambda_{\max}$ (↓) | $\lambda^*$ (↓) | $\mathcal{L}_{\mathrm{LPF}}$ (↓) | $\Delta_{\mathrm{LPF}}$ (↓) |
| Modality $i$: [A] | 10.4196 | 1.3016 | 2.6775 | 0.5937 | Modality $i$: [T] | 8.6738 | 2.0203 | 1.8767 | 1.1044 |
| Modality $j$: [V] | **8.2353** | 1.9430 | 3.2975 | **0.0714** | Modality $j$: [I] | 10.0044 | **1.4935** | 2.9904 | 1.2510 |
| SML | 9.2389 | **1.1461** | 2.5305 | 0.5447 | SML | 8.3238 | 1.8015 | 0.4666 | 0.0937 |
| DML | 9.1712 | 1.6461 | **2.5212** | 0.5112 | DML | **8.3068** | 1.5825 | **0.4353** | **0.0682** |

$\lambda^*$: Ratio of the max $\lambda_{\max}$ (1st) to the 5th largest eigenvalue $\lambda_5$ ($\lambda^* = \lambda_{\max}/\lambda_5$)
$\Delta_{\mathrm{LPF}}$: Discrepancy between original loss and LPF loss.

matched pairs reach to smoother surface.

A slight divergence from the overall trend appears on the CREMA-D dataset. The modality $j$, which is a visual modality, shows marginally smaller values of $\lambda_{\max}$ and $\Delta_{\mathrm{LPF}}$, which would typically indicate a flatter and more robust loss surface. However, this modality still exhibits a relatively larger $\mathcal{L}_{\mathrm{LPF}}$. Moreover, its $\lambda^*$ is highest among all modalities, implying a larger gap between the dominant and subordinate Hessian eigenvalues and thus more curvature. In contrast, both $\lambda_{\mathrm{LPF}}$ and $\lambda^*$ decrease under multimodal learning, indicating that multimodal learning still converges to *flatter minima* relative to the unimodal learning.

### 6.2. Sensitivity to Loss Perturbations

Here, we evaluate the flatness of the loss landscape that has been proposed on SWAD (Cha et al., 2021), $\mathcal{F}_{\mathrm{gap}}(\theta) = \mathbb{E}\left[|\mathcal{E}(\theta') - \mathcal{E}(\theta)|\right]$, which quantifies changes in empirical risk under perturbations to the parameters, and also serves as qualitative results of landscape smoothness in visualization. Here, $\mathcal{E}(\theta)$ denotes the empirical risk at the original parameters $\theta$, and $\mathcal{E}(\theta')$ denotes the risk at perturbed parameters $\theta'$, where $\|\theta'\| = \|\theta\| + \epsilon$ and $\epsilon$ is a small sampled perturbation. A lower $\mathcal{F}_{\mathrm{gap}}(\theta)$ indicates a flatter loss landscape of $\theta$. We sampled 100 times per radius, estimating via Monte Carlo approximation (Metropolis & Ulam, 1949).

As shown in Figure 4, models trained with multimodal data consistently exhibit lower perturbation sensitivity compared to unimodal learning, indicating a smoother optimization. Notably, our suggested DML shows the lowest $\mathcal{F}_{\mathrm{gap}}$ tendency across settings, demonstrating flatter loss landscape.

### 6.3. Feature Variance and Frequency-Domain Analysis

This section provides empirical evidence for the smoothing mechanism described in Theorem 2. We examine two complementary quantities: variance of feature vectors, which serves as the approximation for the effective dispersion of

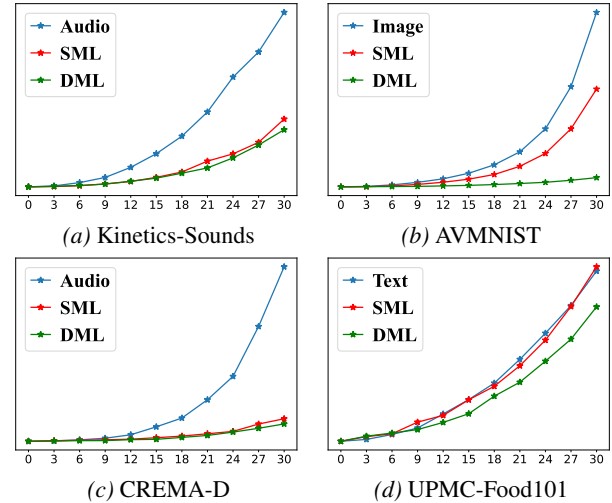

*(a)* Kinetics-Sounds  *(b)* AVMNIST

*(c)* CREMA-D  *(d)* UPMC-Food101

*Figure 4.* **Flatness Comparison between Modality.** These results shows that DML mostly *yields improved flatness compared to single-modality training,* In each plot, the $x$-axis denotes the perturbation radius and the $y$-axis represents $\mathcal{F}_{\mathrm{gap}}(\theta)$.

the scaled-shift kernel $\mathcal{K}_{\mathbf{x}_j, \boldsymbol{\alpha}}(\boldsymbol{\tau})$, and the High-Frequency (HF) Power Ratio, which measures the relative amount of high-frequency spectral energy in the loss landscape.

**Feature Variance** To examine whether DML increases the variance induced by sampling from $p(\mathbf{x}_j \mid y)$, we compare the feature space variance under SML and DML. To evaluate the variance of the feature space empirically, we sampled $M$ data points, where $M$ equals the number of training data. We then construct the concatenated multimodal feature representation $\mathbf{z}_m = [f_i(\mathbf{x}_{i,m}), f_j(\mathbf{x}_{j,m})] \in \mathbb{R}^d$. Given $M$ samples, the feature variance is then computed as

$$\sigma_{\mathbf{z}}^2 = \mathrm{Tr}\left(\frac{1}{dM}\sum_{m=1}^{M}(\mathbf{z}_m - \bar{\mathbf{z}})(\mathbf{z}_m - \bar{\mathbf{z}})^\top\right). \quad (10)$$

Here, $\bar{\mathbf{z}} = \frac{1}{M}\sum_{m=1}^{M}\mathbf{z}_m$ denotes the feature-wise sample mean, and $d$ is the feature dimension. This metric corresponds to the mean dimension-wise unbiased sample vari-

*Table 7.* Feature variance $\sigma_{\mathbf{z}}^2$ under SML and DML across datasets.

|  | KS | AVMNIST | CREMA-D | U.F-101 |
|---|---|---|---|---|
| SML | 0.161 | 0.055 | 0.044 | 0.724 |
| **DML** | **0.200** | **0.079** | **0.048** | **0.733** |

*Table 8.* High-Frequency Power Ratio (%) of the loss landscape.

|  | KS | AVMNIST | CREMA-D | U.F-101 |
|---|---|---|---|---|
| Modality $i$ | 3.70 | 11.42 | 19.67 | 3.38 |
| Modality $j$ | 4.64 | 12.22 | 11.71 | 3.70 |
| SML | 2.89 | 8.01 | 7.84 | 2.10 |
| **DML** | **1.98** | **4.47** | **7.01** | **0.22** |

ance and provides an empirical proxy for the effective variance of the induced smoothing kernel $\mathcal{K}_{\mathbf{x}_j, \boldsymbol{\alpha}}(\boldsymbol{\tau})$.

**High-Frequency Analysis** Motivated by Theorem 2, we examine whether DML suppresses high-frequency components of the loss landscape. We sampled $n$ times of one directional loss curve along layer-wise filter normalization directions in parameter space (Li et al., 2018), and applied Fast Fourier Transform (FFT).

For the detail, let $\ell_n = \mathcal{L}(\boldsymbol{\theta}^\star + \alpha_n \mathbf{d})$ be the sampled loss value at the $n$-th point along the random direction $\mathbf{d}$. With the FFT coefficient $\hat{\ell}_k = \sum_{n=0}^{N-1} \ell_n \exp(-2\pi i k n / N)$, we define the High-Frequency (HF) Power Ratio as

$$\mathcal{K}_{\mathrm{HF}} = \{k \in \mathcal{K}_+ : k \geq 1 + \lfloor \rho | \mathcal{K}_+ | \rfloor\}, \quad (11)$$

$$\mathcal{R}_{\mathrm{HF}} = 100 \times \frac{\sum_{k \in \mathcal{K}_{\mathrm{HF}}} |\hat{\ell}_k|^2}{\sum_{k \in \mathcal{K}_+} |\hat{\ell}_k|^2}, \quad (12)$$

where $\mathcal{K}_+ = \{1, \ldots, \lfloor N/2 \rfloor - 1\}$ excludes DC component and $\rho = 0.2$ denotes low-frequency cutoff ratio, where above this ratio is regarded as high-frequency components.

**Results** The results align with our hypothesis that DML induces a larger-variance smoothing effect and suppresses high-frequency fluctuations in the loss landscape. As shown in Table 7, DML consistently yields higher feature variance than SML across all datasets This indicates that DML induces broader feature-space dispersion, partially reflecting the larger variance of the scaled-shift kernel $\mathcal{K}_{\mathbf{x}_j, \boldsymbol{\alpha}}(\boldsymbol{\tau})$.

Moreover, Table 8 further shows that DML achieves the lowest HF Power Ratio across all datasets, outperforming both unimodal baselines and SML. Since high-frequency components correspond to rapidly varying loss fluctuations along the parameter-space direction, this indicates that DML suppresses high-frequency spectral energy and produces a smoother loss profile. Together, these results support our hypothesis that DML increases the variance of the induced kernel and consequently attenuates high-frequency components of the loss landscape, consistent with Theorem 2.

## 7. Discussion

In this section, we discuss the validity of the conditional independence assumption in Equation (2), particularly under conditioning on the output label $y$. Furthermore, we then address whether SML and DML induce theoretically identical underlying distributions under this condition.

**The Role of $y$ in Multimodal Learning** The goal of multimodal learning is not to model incidental correlations between modalities, but to learn a mapping from input space to the underlying output space. Under this perspective, Equation (2) characterizes reasonable setting: once the true $y$ is specified, individual modalities are not required to be highly correlated with each other; rather, they serve as independent evidence supporting that hypothesis. This viewpoint is consistent with a wide range of multimodal tasks in which modalities serve as complementary but conditionally independent with given target output $y$, whether $y$ corresponds to a label space or a semantic space. Empirically, our experiments across diverse tasks, such as multimodal classification (Table 2) and Image-Text retrieval (Table 5), demonstrate that this DML setting is more effective.

**Sampling Space: SML vs. DML** The key distinction between SML and DML lies not in the conditional distributions they rely on, but in the effective sampling space they cover. Under Equation (2), DML samples from the full conditional product space, *i.e.*, $p(\mathbf{x}_i, \mathbf{x}_j \mid y) = p(\mathbf{x}_i \mid y)p(\mathbf{x}_j \mid y)$, and therefore covers all valid cross-modal pairings consistent with the output label $y$. By contrast, SML is restricted to the finite paired set observed in the original dataset, excluding many valid cross-modal combinations that do not appear as raw pairs. Thus, although SML and DML are grounded in the same conditional marginals, SML explores only a narrow observed subset, whereas DML covers a much broader space of valid cross-modal pairings. In this sense, SML is a restricted special case of DML, and DML substantially enlarges the effective sampling space without changing the underlying conditional marginals.

## 8. Conclusion

In this paper, we analyze multimodal learning through the lens of the loss landscape. Our theoretical viewpoints suggest that integrating multiple modalities induces a convolutional smoothing effect, leading to a smoother loss surface. Building on this insight, we propose a strategy that stochastically matches same-target $y$ samples across modalities in the shared output space $\mathcal{Y}$, forecasting distributional learning and implicitly regularizing the hypothesis space $h_\theta : \mathcal{X} \to \mathcal{Y}$ toward a smoother loss landscape. Finally, we outline future research directions, including scalability issues with models and datasets, and extensions to diverse tasks, which are essential for multimodal applications.

## Acknowledgments

This work was supported by the National Research Foundation of Korea (NRF) grant funded by the Korea government (MSIT) (No. RS-2024-00459023), the Institute of Information & Communications Technology Planning & Evaluation (IITP) grant funded by the Korean Government (MSIT): (No. RS-2020-II201336, Artificial Intelligence Graduate School Program at UNIST), (No. RS-2025-25442824, AI Star Fellowship Program (UNIST)), (No. IITP-2026-RS-2022-00156361, Innovative Human Resource Development for Local Intellectualization Program), (No. RS-2026-25528781, Hyper-scale Industrial AI Research Support (R&D) Program, Development of an industry-specified intelligent data processing and federated learning platform).

## Impact Statement

This paper presents the method designed to improve the optimization landscape and robustness capability of multimodal learning. By leveraging stochastic pairing to smooth the loss landscape, our work contributes to the development of more robust and reliable AI systems in this field. Improving the stability of multimodal representations is critical for deploying machine learning models in real-world scenarios where data may be noisy or diverse. Then our works can contribute enhancing the fundamental training approach of multimodal models serves as a step toward more stable and trustworty AI applications We do not foresee immediate negative societal consequences.

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

# A. Mathematical Proofs

We first recall our notations and the definitions:

**Notations:**

- $\mathbf{x}_i \in \mathcal{X}_i \subset \mathbb{R}^{d_i}, \mathbf{x}_j \in \mathcal{X}_j \subset \mathbb{R}^{d_j}$ denote input data from two different modalities $i$ and $j$, with their data distribution $\mathbf{x}_i \sim p(\mathbf{x}_i)$ and $\mathbf{x}_j \sim p(\mathbf{x}_j)$, respectively.

- $y \in \mathcal{Y}$ be the target label, which indicates the shared semantics across modalities.

- $f_i(\mathbf{x}_i), f_j(\mathbf{x}_j) \in \mathbb{R}^d$ are the modality encoders for the respective modality.

- $f(\mathbf{x}_i, \mathbf{x}_j; \theta) := \phi(f_i(\mathbf{x}_i), f_j(\mathbf{x}_j))$ be a function parametrized by $\theta \in \mathbb{R}^m$, with modality specific function $f_i(\cdot)$ and $f_j(\cdot)$, and modality fusion function $\phi(\cdot, \cdot)$

- $\ell(f(\mathbf{x}_i, \mathbf{x}_j; \theta), y)$ is the loss function.

- Equation (3): $\mathcal{L}_{\text{uni}} = \underset{y \in \mathcal{Y}}{\mathbb{E}} \left[ \int \ell(f(\mathbf{x}_i, \bar{\mathbf{x}}_j; \theta), y) p(\mathbf{x}_i \mid y) \, d\mathbf{x}_i \right]$

- Equation (4): $\mathcal{L}_{\text{multi}} = \underset{y \in \mathcal{Y}}{\mathbb{E}} \left[ \iint \ell(f(\mathbf{x}_i, \mathbf{x}_j; \theta), y) p(\mathbf{x}_i, \mathbf{x}_j \mid y) \, d\mathbf{x}_i \, d\mathbf{x}_j \right]$

**Definition 4.1**(**Fusion-Induced Scaled Shift and Kernel**) *Let $\bar{\mathbf{x}}_j$ denote input data representing absence of modality $j$, such as zero vector or constant value, and define fusion-induced scaled shift as :*

$$\tau_{\boldsymbol{\alpha}}(\mathbf{x}_j) := \boldsymbol{\alpha} \odot \left( \phi(f_i(\mathbf{x}_i), f_j(\mathbf{x}_j)) - \phi(f_i(\mathbf{x}_i), f_j(\bar{\mathbf{x}}_j)) \right), \tag{13}$$

*The corresponding* scaled-shift kernel *is then defined using the Dirac delta function* $\delta_{\tau_{\boldsymbol{\alpha}}(\mathbf{x}_j)}(\cdot)$ *centered at* $\tau_{\boldsymbol{\alpha}}(\mathbf{x}_j)$:

$$\mathcal{K}_{\mathbf{x}_j, \boldsymbol{\alpha}}(\boldsymbol{\tau}) := \underset{\mathbf{x}_j \sim p(\mathbf{x}_j | y)}{\mathbb{E}} \left[ \delta_{\tau_{\boldsymbol{\alpha}}(\mathbf{x}_j)}(\boldsymbol{\tau}) \right], \tag{14}$$

## A.1. Proofs of Theorem 1

Before the proof, we recall our Theorem 1:

**Theorem 1** (**Convolutional Smoothing with Modality Scaled Shifted Kernel**) *The expected multimodal loss is a scaled convolution of the unimodal loss:*

$$\mathcal{L}_{\text{multi}} = \underset{y \in \mathcal{Y}}{\mathbb{E}} \left[ \left( \mathcal{L}_{\text{uni}} \circledast \mathcal{K}_{\mathbf{x}_j, \boldsymbol{\alpha}} \right) (f \mid y) \right]. \tag{15}$$

*Proof.* According to Equation (4), we express the multimodal loss in terms of $\tau_{\boldsymbol{\alpha}}(\mathbf{x}_j)$ from Definition 1:

$$\mathcal{L}_{\text{multi}}(\cdot \mid y) = \iint \ell\left( \phi(f_i(\mathbf{x}_i), f_j(\mathbf{x}_j)), y \right) p(\mathbf{x}_i, \mathbf{x}_j \mid y) \, d\mathbf{x}_i \, d\mathbf{x}_j \tag{16}$$

$$= \iint \ell\left( \phi(f_i(\mathbf{x}_i), f_j(\bar{\mathbf{x}}_j)) + \tau_{\boldsymbol{\alpha}}(\mathbf{x}_j), y \right) p(\mathbf{x}_i, \mathbf{x}_j \mid y) \, d\mathbf{x}_i \, d\mathbf{x}_j \tag{17}$$

$$= \iint \ell\left( \phi(f_i(\mathbf{x}_i), f_j(\bar{\mathbf{x}}_j)) + \tau_{\boldsymbol{\alpha}}(\mathbf{x}_j), y \right) p(\mathbf{x}_i \mid y) p(\mathbf{x}_j \mid y) \, d\mathbf{x}_i \, d\mathbf{x}_j, \tag{18}$$

Let $f := \phi(f_i(\mathbf{x}_i), f_j(\bar{\mathbf{x}}_j)) \in \mathbb{R}^d$. Then the above becomes

$$\mathcal{L}_{\text{multi}}(\cdot \mid y) = \int \left[ \int \ell\left( f + \tau_{\boldsymbol{\alpha}}(\mathbf{x}_j), y \right) p(\mathbf{x}_j \mid y) \, d\mathbf{x}_j \right] p(\mathbf{x}_i \mid y) \, d\mathbf{x}_i. \tag{19}$$

By Definition 1, we set the scaled-shift kernel

$$\mathcal{K}_{\mathbf{x}_j, \boldsymbol{\alpha}}(\boldsymbol{\tau}) := \underset{\mathbf{x}_j \sim p(\mathbf{x}_j | y)}{\mathbb{E}} \left[ \delta_{\tau_{\boldsymbol{\alpha}}(\mathbf{x}_j)}(\boldsymbol{\tau}) \right], \tag{20}$$

where $\delta_{\tau_{\boldsymbol{\alpha}}(\mathbf{x}_j)}$ is the Dirac delta centered at $\tau_{\boldsymbol{\alpha}}(\mathbf{x}_j)$. The inner integral over $\mathbf{x}_j$ is therefore the integral of $\ell(f + \boldsymbol{\tau}, y)$ against this kernel, *i.e.*

$$\int \ell(f + \boldsymbol{\tau}, y) \, \mathcal{K}_{\mathbf{x}_j, \boldsymbol{\alpha}}(\boldsymbol{\tau}) \, d\boldsymbol{\tau} = \big(\ell(\cdot, y) \circledast \mathcal{K}_{\mathbf{x}_j, \boldsymbol{\alpha}}\big)(f). \tag{21}$$

Hence the conditional multimodal loss is the convolution of the unimodal loss with the scaled-shift kernel. Taking expectation over $y \in \mathcal{Y}$ completes the proof:

$$\mathcal{L}_{\text{multi}} = \mathbb{E}_{y \in \mathcal{Y}} \big[\big(\mathcal{L}_{\text{uni}} \circledast \mathcal{K}_{\mathbf{x}_j, \boldsymbol{\alpha}}\big)(f \mid y)\big]. \tag{22}$$

$\square$

**Theoretical Justification of the $\mathbf{x}_j$** The smoothing effect described in Theorem 1 critically depends on the auxiliary modality $\mathbf{x}_j$ containing label-dependent information. If $\mathbf{x}_j$ is statistically independent of the label (*i.e.*, $p(\mathbf{x}_j \mid y) = p(\mathbf{x}_j)$), then the optimal training objective will suppress its influence.

Consider, for example, the degenerate case where $\mathbf{x}_j \sim \mathcal{N}(0, \sigma^2 I)$ is high-variance Gaussian noise independent of $y$. In this case, for each $\mathbf{x}_i$, the model minimizes

$$\min_{\mathbf{x}_j \sim p(\mathbf{x}_j)} \mathbb{E}\big[\ell(f(\mathbf{x}_i, \mathbf{x}_j; \theta), y)\big], \tag{23}$$

and because $\mathbf{x}_j$ carries no label information, the optimal mapping tends to a constant representation $f_j(\mathbf{x}_j) \approx \mathbf{c}$ for some fixed vector $\mathbf{c}$. The fusion-induced shift (Definition 1) then becomes approximately deterministic, producing an induced kernel that collapses to a Dirac delta at the origin:

$$\mathcal{K}_{\mathbf{x}_j, \boldsymbol{\alpha}} = \delta_0(\boldsymbol{\tau}). \tag{24}$$

According to Theorem 1, convolution with a Dirac delta kernel leaves the loss unchanged:

$$\mathcal{L}_{\text{multi}} = \mathcal{L}_{\text{uni}} \circledast \delta_0 = \mathcal{L}_{\text{uni}}. \tag{25}$$

Thus, even though the input noise of $\mathbf{x}_j$ has high variance, the model's training suppresses it entirely, leading to zero variance in the shift kernel and no smoothing effect. This analysis reinforces that meaningful convolutional smoothing—and thus our theoretical results—require the auxiliary modality to be label-dependent. When $\mathbf{x}_j$ is non-informative, the multimodal loss naturally collapses to the unimodal form, confirming the role of label dependence in generating useful smoothing behavior.

### A.2. Proofs of Theorem 2

We also recall our Theorem 2 before the proof:

**Theorem 2 (Loss Landscape of Multimodal Learning)** *Suppose that conditional unimodal loss $\mathcal{L}_{uni}(f \mid y)$ is continuous and bounded for all $y \in \mathcal{Y}$, and the shift kernel $\mathcal{K}_{\mathbf{x}_j, \boldsymbol{\alpha}}$ has finite variance. Then the $\mathcal{L}_{multi}$ is smoother than the $\mathcal{L}_{uni}$, in the following two folds:*

1. ***Upper Bound on the Spectral Norm of the Hessian:*** *For any $f$, the spectral norm of the multimodal loss Hessian is upper-bounded by that of the unimodal loss Hessian:*

$$\sup_f \|\nabla^2 \mathcal{L}_{multi}(f \mid y)\|_2 \leq \sup_f \|\nabla^2 \mathcal{L}_{uni}(f \mid y)\|_2. \tag{26}$$

2. ***Frequency Domain Interpretation:*** *Let $\widehat{\mathcal{L}}_{multi}(\omega)$, $\widehat{\mathcal{L}}_{uni}(\omega)$, and $\widehat{\mathcal{K}}_{\mathbf{x}_j, \boldsymbol{\alpha}}(\omega)$ denote the Fourier transforms of $\mathcal{L}_{multi}$, $\mathcal{L}_{uni}$, and $\mathcal{K}_{\mathbf{x}_j, \boldsymbol{\alpha}}$ respectively. Then, Equation (7) can be expressed in the frequency domain as: $\widehat{\mathcal{L}}_{multi}(\omega) = \widehat{\mathcal{L}}_{uni}(\omega) \cdot \widehat{\mathcal{K}}_{\mathbf{x}_j, \boldsymbol{\alpha}}(\omega)$. Moreover, it follows that:*

$$\mathcal{O}(\widehat{\mathcal{L}}_{multi}(\omega)) \leq \mathcal{O}(\widehat{\mathcal{L}}_{uni}(\omega)). \tag{27}$$

*Proof.* We prove each of the statements in the Theorem 2 as follows:

**1. Spectral Norm Upper Bound on the Multimodal Hessian** We prove that the maximal spectral norm of the Hessian of the multimodal loss is upper-bounded by that of the unimodal loss.

Let the Hessians of the unimodal and multimodal losses at a point $f \in \mathbb{R}^d$ be defined as:

$$H_{\text{uni}}(f \mid y) := \nabla^2 \mathcal{L}_{\text{uni}}(f \mid y), \qquad H_{\text{multi}}(f \mid y) := \nabla^2 \mathcal{L}_{\text{multi}}(f \mid y). \tag{28}$$

From the convolutional smoothing formulation (Equation (7)), the multimodal loss is given by:

$$\mathcal{L}_{\text{multi}}(f \mid y) = \left( \mathcal{L}_{\text{uni}}(\cdot \mid y) \circledast \mathcal{K}_{\mathbf{x}_j, \boldsymbol{\alpha}} \right)(f) \tag{29}$$

$$= \int \mathcal{L}_{\text{uni}}(f + \boldsymbol{\tau} \mid y) \, \mathcal{K}_{\mathbf{x}_j, \boldsymbol{\alpha}}(\boldsymbol{\tau}) \, d\boldsymbol{\tau}, \tag{30}$$

so differentiating twice with respect to $f$ yields

$$H_{\text{multi}}(f \mid y) = \int H_{\text{uni}}(f + \boldsymbol{\tau} \mid y) \, \mathcal{K}_{\mathbf{x}_j, \boldsymbol{\alpha}}(\boldsymbol{\tau}) \, d\boldsymbol{\tau}. \tag{31}$$

Taking the spectral norm on both sides yields:

$$\|H_{\text{multi}}(f \mid y)\|_2 = \left\| \int H_{\text{uni}}(f + \boldsymbol{\tau} \mid y) \, \mathcal{K}_{\mathbf{x}_j, \boldsymbol{\alpha}}(\boldsymbol{\tau}) \, d\boldsymbol{\tau} \right\|_2 \tag{32}$$

$$\leq \int \left\| H_{\text{uni}}(f + \boldsymbol{\tau} \mid y) \right\|_2 \mathcal{K}_{\mathbf{x}_j, \boldsymbol{\alpha}}(\boldsymbol{\tau}) \, d\boldsymbol{\tau}. \tag{33}$$

where the inequality follows from Jensen's inequality, given that the spectral norm $\| \cdot \|_2$ is a convex function over symmetric matrices. Now, we take the supremum over $f \in \mathbb{R}^d$. For every fixed $\boldsymbol{\tau}$ the map $f \mapsto f + \boldsymbol{\tau}$ is a bijection of $\mathbb{R}^d$ with additional notation $u = f + \boldsymbol{\tau}$, hence

$$\sup_f \left\| H_{\text{uni}}(f + \boldsymbol{\tau} \mid y) \right\|_2 = \sup_u \left\| H_{\text{uni}}(u \mid y) \right\|_2, \tag{34}$$

Then, we derive the bound of the largest spectral norm of the Hessian of multimodal learning relative to unimodal learning as follows:

$$\sup_f \|H_{\text{multi}}(f \mid y)\|_2 \leq \sup_f \int \left\| H_{\text{uni}}(f + \boldsymbol{\tau} \mid y) \right\|_2 \mathcal{K}_{\mathbf{x}_j, \boldsymbol{\alpha}}(\boldsymbol{\tau}) \, d\boldsymbol{\tau} \tag{35}$$

$$\leq \int \left( \sup_u \left\| H_{\text{uni}}(u \mid y) \right\|_2 \right) \mathcal{K}_{\mathbf{x}_j, \boldsymbol{\alpha}}(\boldsymbol{\tau}) \, d\boldsymbol{\tau} \tag{36}$$

$$= \sup_u \left\| H_{\text{uni}}(u \mid y) \right\|_2 \cdot \int \mathcal{K}_{\mathbf{x}_j, \boldsymbol{\alpha}}(\boldsymbol{\tau}) \, d\boldsymbol{\tau} \tag{37}$$

$$= \sup_f \left\| H_{\text{uni}}(f \mid y) \right\|_2 \tag{38}$$

The final equality holds for two reasons. First, $\mathcal{K}_{\mathbf{x}_j, \boldsymbol{\alpha}}$ is a probability density function, and this satisfies $\int \mathcal{K}_{\mathbf{x}_j, \boldsymbol{\alpha}}(\boldsymbol{\tau}) \, d\boldsymbol{\tau} = 1$. Second, because the supremum is taken over all of $\mathbb{R}^d$, the variable is a dummy variable, and this means it is valid to rename $u \mapsto f$.

Finally, the supremum of the Hessian of the multimodal loss is bounded by that of the unimodal loss,

$$\sup_f \left\| \nabla^2 \mathcal{L}_{\text{multi}}(f \mid y) \right\|_2 \leq \sup_f \left\| \nabla^2 \mathcal{L}_{\text{uni}}(f \mid y) \right\|_2, \tag{39}$$

This result implies that $\mathcal{L}_{\text{multi}}$ landscape, in terms of its maximal curvature (*i.e.*, the largest spectral norm of the Hessian), is globally smoother or at most equally sharp as that of the $\mathcal{L}_{\text{uni}}$.

**2. Frequency Domain Interpretation** Consider loss function $\mathcal{L}(f(\cdot, \cdot; \theta))$, where $f$ is parameterized by $\theta \in \mathbb{R}^m$. We treat loss as a function over the parameter space $\theta$, and analyze its behavior in the frequency domain. We define the Fourier transform of the loss function with respect to $\theta$ as:

$$\widehat{\mathcal{L}}(\omega) = \int \mathcal{L}(f(\cdot, \cdot; \theta)) \, e^{-i \omega^\top \theta} \, d\theta, \quad \omega \in \mathbb{R}^m. \tag{40}$$

By the convolution theorem in Fourier analysis, the Fourier transform of the convolution of two functions equals the pointwise product of their individual Fourier transforms:

$$\mathcal{F}[f \circledast g](\omega) = \mathcal{F}[f](\omega) \cdot \mathcal{F}[g](\omega). \tag{41}$$

Applying this to Equation (7), which expresses the $\mathcal{L}_{\mathrm{multi}}$ as a convolution over parameter space, we obtain:

$$\widehat{\mathcal{L}}_{\mathrm{multi}}(\omega) = \widehat{\mathcal{L}}_{\mathrm{uni}}(\omega) \cdot \widehat{\mathcal{K}}_{\mathbf{x}_j, \boldsymbol{\alpha}}(\omega), \tag{42}$$

where $\widehat{\mathcal{K}}_{\mathbf{x}_j, \boldsymbol{\alpha}}(\omega)$ denotes the Fourier transform of the shift kernel applied in the parameter space $\theta$.

To analyze the asymptotic behavior, we consider the decay properties of $\widehat{\mathcal{L}}_{\mathrm{multi}}(\omega)$ and $\widehat{\mathcal{L}}_{\mathrm{uni}}(\omega)$ as $\|\omega\| \to \infty$. Moreover, this consideration shows that the decay rate of $\widehat{\mathcal{K}}_{\mathbf{x}_j, \boldsymbol{\alpha}}(\omega)$ decides how much high-frequency content is preserved or attenuated by the $\mathcal{L}_{\mathrm{multi}}$. Since the shift kernel is defined as: $\mathcal{K}_{\mathbf{x}_j, \boldsymbol{\alpha}}(\boldsymbol{\tau}) := \mathbb{E}_{\mathbf{x}_j \sim p(\mathbf{x}_j | y)} \left[ \delta_{\tau_{\boldsymbol{\alpha}}(\mathbf{x}_j)}(\boldsymbol{\tau}) \right]$, which is the expectation of a Dirac delta centered at $\tau_{\boldsymbol{\alpha}}(\mathbf{x}_j)$, then its Fourier transform is the *characteristic function* of the random variable $\tau_{\boldsymbol{\alpha}}(\mathbf{x}_j)$ as follows:

$$\widehat{\mathcal{K}}_{\mathbf{x}_j, \boldsymbol{\alpha}}(\omega) = \mathbb{E}_{\mathbf{x}_j \sim p(\mathbf{x}_j | y)} \left[ e^{-i\omega^\top \tau_{\boldsymbol{\alpha}}(\mathbf{x}_j)} \right]. \tag{43}$$

As we assumed that $\tau_{\boldsymbol{\alpha}}(\mathbf{x}_j)$ has finite variance, this characteristic function is continuous and satisfies the following properties (Körner, 1988; Billingsley, 1995):

$$\left| \widehat{\mathcal{K}}_{\mathbf{x}_j, \boldsymbol{\alpha}}(\omega) \right| \leq 1, \quad \text{and} \quad \lim_{\|\omega\| \to \infty} \left| \widehat{\mathcal{K}}_{\mathbf{x}_j, \boldsymbol{\alpha}}(\omega) \right| = 0. \tag{44}$$

Consequently, the magnitude of $\mathcal{L}_{\mathrm{multi}}$ in the frequency domain is bounded above by that of the $\mathcal{L}_{\mathrm{uni}}$:

$$\left| \widehat{\mathcal{L}}_{\mathrm{multi}}(\omega) \right| = \left| \widehat{\mathcal{L}}_{\mathrm{uni}}(\omega) \right| \cdot \left| \widehat{\mathcal{K}}_{\mathbf{x}_j, \boldsymbol{\alpha}}(\omega) \right| \leq \left| \widehat{\mathcal{L}}_{\mathrm{uni}}(\omega) \right| \tag{45}$$

This implies that the high-frequency components of the $\mathcal{L}_{\mathrm{multi}}$ are effectively suppressed relative to those of the $\mathcal{L}_{\mathrm{uni}}$. Hence, in terms of asymptotic decay:

$$\mathcal{O}\left( \widehat{\mathcal{L}}_{\mathrm{multi}}(\omega) \right) \leq \mathcal{O}\left( \widehat{\mathcal{L}}_{\mathrm{uni}}(\omega) \right). \tag{46}$$

This frequency-domain analysis highlights the smoothing effect induced by the $\mathcal{L}_{\mathrm{multi}}$ formulation, which naturally dampens high-frequency variations in the parameter space $\theta \in \mathbb{R}^m$. $\qquad \square$

### A.3. Empirical Validation of Approximate Translation Invariance (ATI)

**Applicability of ATI to Fusion Networks** Our theoretical analysis focuses on late-fusion architectures, which are widely used in multimodal learning. We adopt the approximate translation-invariance (ATI) property (Equation (1)), which postulates that the fusion network exhibits near-invariant behavior under small shifts in one modality. We analyze four widely used fusion mechanisms: Additive Fusion, Concatenation with linear MLP, Concatenation with non-linear MLP, and Cross-Attention (on infinite dimension) to determine the extent to which satisfies the ATI property:

- **Additive Fusion**: In this case, the assumption holds exactly with $\boldsymbol{\alpha} = \mathbf{1}$, as

$$\phi(\mathbf{u}, \mathbf{v} + \boldsymbol{\tau}) = \phi(\mathbf{u}, \mathbf{v}) + \boldsymbol{\tau} \tag{47}$$

- **Concatenation + Linear MLP**: Let $\phi(\mathbf{u}, \mathbf{v}) = W[\mathbf{u} \mid \mathbf{v}] + \mathbf{b}$, where the weights are decomposed as $W = [W_{\mathbf{u}} \mid W_{\mathbf{v}}]$. When a small perturbation $\boldsymbol{\tau}$ is applied to modality $\mathbf{v}$ (assuming the input dimension of $\mathbf{v}$ matches the output dimension $d$), the perturbed output is given exactly by:

$$\phi(\mathbf{u}, \mathbf{v} + \boldsymbol{\tau}) = \phi(\mathbf{u}, \mathbf{v}) + W_{\mathbf{v}} \boldsymbol{\tau}. \tag{48}$$

  While the transformation relies on the full dense matrix $W_{\mathbf{v}}$, the ATI property characterizes the interaction via element-wise modulation. By isolating the diagonal as $\boldsymbol{\alpha} := \mathrm{diag}(W_{\mathbf{v}})$, we can approximate this linear effect as:

$$\phi(\mathbf{u}, \mathbf{v} + \boldsymbol{\tau}) \approx \phi(\mathbf{u}, \mathbf{v}) + \boldsymbol{\alpha} \odot \boldsymbol{\tau}. \tag{49}$$

- **Concatenation + Non-linear MLP**: For deeper nonlinear networks (*e.g.*, including ReLU activations), the ATI property holds approximately through a first-order Taylor expansion:

$$\phi(\mathbf{u}, \mathbf{v} + \boldsymbol{\tau}) \approx \phi(\mathbf{u}, \mathbf{v}) + J_\phi \cdot \boldsymbol{\tau} + R(\|\boldsymbol{\tau}\|^2) \tag{50}$$

where $J_\phi := \frac{\partial \phi}{\partial \mathbf{v}} \in \mathbb{R}^{d \times d_j}$ is the Jacobian of $\phi$ with respect to $\mathbf{v}$. Moreover, $R(\|\boldsymbol{\tau}\|^2)$ is the higher-order remainder term satisfying $\mathcal{O}(\|\boldsymbol{\tau}\|^2)$ as $\|\boldsymbol{\tau}\|^2 \to 0$. This expansion shows that, under mild smoothness assumptions on $\phi$, the output shift is approximately linear in $\boldsymbol{\tau}$, and thus the ATI property: $\phi(\mathbf{u}, \mathbf{v} + \boldsymbol{\tau}) \approx \phi(\mathbf{u}, \mathbf{v}) + \boldsymbol{\alpha} \odot \boldsymbol{\tau}$, holds approximately if we let $\boldsymbol{\alpha} := \mathrm{diag}(J_\phi)$, *i.e.*, the diagonal of the Jacobian. It provides an approximation to the shift effect that aligns with the elementwise form of the ATI property.

- **Cross-Attention.** Let $\mathbf{u}$ provides the queries, while $\mathbf{v}$ provides the keys and values within cross-attention block:

$$\phi(\mathbf{u}, \mathbf{v}) = \mathrm{Attn}(Q\mathbf{u}, K\mathbf{v}, V\mathbf{v}) \tag{51}$$

$$= \mathrm{softmax}\left(\frac{(Q\mathbf{u})(K\mathbf{v})^\top}{\sqrt{d_k}}\right) V\mathbf{v}. \tag{52}$$

Now suppose that a perturbation $\boldsymbol{\tau}$ is applied to $\mathbf{v}$. Since $\mathbf{v}$ is used to construct both keys and values, this perturbation induces $K(\mathbf{v} + \boldsymbol{\tau}) = K\mathbf{v} + K\boldsymbol{\tau}$ and $V(\mathbf{v} + \boldsymbol{\tau}) = V\mathbf{v} + V\boldsymbol{\tau}$. Therefore, $\boldsymbol{\tau}$ affects the cross-attention output through two pathways: it changes the attention weights through the keys and changes the attended features through the values.

Then, we can locally linearize the perturbed cross-attention output around $\mathbf{v}$. A first-order Taylor expansion gives

$$\phi(\mathbf{u}, \mathbf{v} + \boldsymbol{\tau}) = \phi(\mathbf{u}, \mathbf{v}) + J_\phi \boldsymbol{\tau} + R(\|\boldsymbol{\tau}\|^2), \tag{53}$$

where $R(\|\boldsymbol{\tau}\|^2)$ denotes the higher-order remainder, and

$$J_\phi := \frac{\partial \phi}{\partial (K\mathbf{v})} K + \frac{\partial \phi}{\partial (V\mathbf{v})} V \in \mathbb{R}^{d \times d_j} \tag{54}$$

is the Jacobian of the cross-attention output with respect to $\mathbf{v}$. The first term captures the sensitivity of the attention weights to perturbations in the keys, while the second term captures the direct sensitivity of the attended values. This linearization is used as a local, module-level analytical lens for the cross-attention block, characterizing how a small perturbation in one modality propagates through attention. Related NTK- and lazy-training-style analyses (Lee et al., 2019; Jacot et al., 2018; Yang, 2020) motivate that, in sufficiently wide networks and locally around initialization, the attention Jacobian may vary slowly. Thus, we obtain the local approximation $\phi(\mathbf{u}, \mathbf{v} + \boldsymbol{\tau}) \approx \phi(\mathbf{u}, \mathbf{v}) + J_\phi \boldsymbol{\tau}$.

These derivations indicate that our ATI property is consistent with the behavior of recent complex models under the standard theoretical framework of modern deep learning. Therefore, based on Table 1 and above mathematical derivations, these findings provide theoretical framework that satisfies linearization and empirical validation across a wide range of practical architectures, and the approximation is still expected to become increasingly accurate in the infinite-dimensional regime.

**Empirical Validity of ATI Property** We provide additional empirical evidence supporting the validity of Approximate Translation Invariance (ATI), beyond the linearization analysis in the main paper. Building on Table 1, we further evaluate a deeper cross-attention fusion module with 8 attention heads and 4 input tokens. As shown in Table 9, the relative approximation error remains small as the number of attention layers increases. This result indicates that most fusion modules generally satisfies first-order local approximation.

*Table 9.* Number of Attention. Layers

| # Layers | 1 | 2 | 4 |
|---|---|---|---|
| $\epsilon_{rel} \times 100$ | 1.20 | 1.22 | 1.28 |

# B. Additional Experimental Details

## B.1. Dataset Descriptions

**Kinetics-Sounds** Kinetics-Sounds (Arandjelovic & Zisserman, 2017) is a multimodal dataset that comprises both audio and visual modalities, derived as a subset of the Kinetics-400 dataset (Carreira & Zisserman, 2017). The Kinetics-Sounds contains 34 human action classes with . Each clip has an approximate duration of 10 seconds and is sourced from distinct YouTube videos. For the visual modality preprocessing, we extract video frames at a rate of 1 frame per second, with each frame resized to $224 \times 224 \times 3$. During iterative training, we employ a random sampling strategy where 3 frames are

selected per class, resulting in a total sampling of batch size $\times$ 1. The audio modality contains sound clips from the same videos, which are converted into mel-spectrograms with a resolution of $128 \times 128 \times 1$. We split training set with a 9:1 ratio to create validation set.

**AVMNIST** AVMNIST (Vielzeuf et al., 2018) is a multimodal dataset comprising audio and visual modalities. The visual modality consists of digit images (0–9) from MNIST (Lecun et al., 1998), with each image having a resolution of $28 \times 28 \times 1$. These images is PCA-projected, retaining 75% of the energy. The audio modality includes spoken digit recordings from the Free Spoken Digit Dataset (FSDD) (Jackson et al., 2018). The audio samples are preprocessed into mel-spectrograms, each with a resolution of $112 \times 112 \times 1$. We split training datasets into 9:1 ratio to create validation datasets.

**CREMA-D** CREMA-D (Cao et al., 2014) is an audio-visual modality dataset designed for speech emotion recognition, consisting of 7,442 video clips from 91 actors (48 male and 43 female). The dataset includes six emotions: *Anger, Disgust, Fear, Happy, Neutral, and Sad.* For the visual modality, we pre-process 1 frame per second of each video. During training, we randomly sample one frame per class, resulting in an overall image batch size of (batch size $\times$ 1). Audio data is preprocessed by resampling into 22,050 Hz and converted into spectrograms using the Short-Time Fourier Transform (STFT) with a 512-point FFT and a hop length of 353, and then log-scale magnitude of spectrograms. Following prior works (Peng et al., 2022; Li et al., 2023), we followed the same dataset splitting strategy. As the original dataset provides only training and test datasets, we further randomly split the training dataset into a 9:1 ratio to create a validation split.

**UPMC-Food101** UPMC-Food101 (Wang et al., 2015) is a large-scale multimodal dataset comprising paired text and image modalities, specifically curated for food recognition and recipe understanding tasks. The dataset encompasses 101 different food categories with corresponding recipes, with over 100,000 items collected from the web. For preprocessing the visual modality, all food images are standardized by resizing to $224 \times 224 \times 3$. For the text modality, which contains recipe descriptions and ingredient lists, we use the BERT model (Devlin et al., 2019) along with its corresponding tokenizer, a de facto choice for encoder-based transformer architectures in recent studies. Since a validation set is not provided, we split the original training set using a 9:1 ratio to create one.

### B.2. Implementation Detail and Hyperparameter

**Computational Resources** We utilized the configuration of computational resources and frameworks as follows:

- **CPU**: Intel(R) Xeon(R) Gold 6342 @ 2.80GHz with 256GB RAM

- **GPU**: Single NVIDIA RTX A5000 with 24GB VRAM

- **Deep Learning Framework**: PyTorch 2.0.1 with CUDA 11.8

- **Codebase**: Adopted official implementation of Adaptive Gradient Modulation (Li et al., 2023)

**Hyperparameter Details** We provide experimental details for all datasets in Table 10. We mostly modified the configuration that has been provided in BalanceBenchmark (Xu et al., 2025).

*Table 10.* **Hyperparameter Settings**

| Settings | Kinetics-Sounds | AVMNIST | CREMA-D | UPMC-Food101 |
|---|---|---|---|---|
| Epochs | 70 | 70 | 70 | 70 |
| Batch Size | 64 | 64 | 16 | 64 |
| Learning Rate (LR) | 0.001 | 0.001 | 0.001 | 0.001 (text) / 0.01 (image) |
| Weight Decay | 0.0005 | 0.0005 | 0.0005 | 0.001 (text) / 0.0001 (image) |
| Optimizer | Adam | SGD | SGD | SGD |
| Scheduler | StepLR | StepLR | StepLR | StepLR |
| Decay Step / Ratio | 30 / 0.1 | 30 / 0.1 | 50 / 0.1 | 30 / 0.1 |
| Model Architecture | ResNet | ResNet | ResNet | (Pretrain) BERT & ResNet-18 |

**Hyperparameters for Retrieval Task** Additionally, for DML approach for Image-Text Retrieval tasks, we fine-tuned the model using a learning rate of $1 \times 10^{-5}$ , a weight decay of 0.01, a batch size of 256, and gradient clipping with a norm of 1.0. The model was trained for a total of 10 epochs, with a warm-up phase applied over the first 10% of the total training iterations. This experiment has conducted on a single GPU with 24GB VRAM. As mentioned previously, we used $K = 1000$ for $K$-Means clustering.

---

**Algorithm 1** Distributional Multimodal Learning (DML)

---

**Require:** Labeled paired training set $\mathcal{D} = \{(\mathbf{x}_{i,n}, \mathbf{x}_{j,n}, y_n)\}_{n=1}^{N}$, multimodal model $f(\mathbf{x}_i, \mathbf{x}_j; \theta)$, loss $\ell(f(\cdot; \theta), y)$, learning rate $\eta$

**Ensure:** Trained parameters $\theta$

1: Initialize model parameters $\theta$
2: **while** not converged **do**
3:     Sample a mini-batch $\mathcal{B} \subset \{1, \ldots, N\}$
4:     **for** each anchor index $b \in \mathcal{B}$ **do**
5:         Construct same-label candidate set $\mathcal{I}_j(y_b) = \{k \in \{1, \ldots, N\} : y_k = y_b\}$
6:         Sample $k_b \sim \text{Uniform}(\mathcal{I}_j(y_b))$, corresponding to $\mathbf{x}_{j,k_b} \sim p(\mathbf{x}_j \mid y_b)$
7:         Form a DML pair $(\mathbf{x}_{i,b}, \mathbf{x}_{j,k_b}, y_b)$
8:     **end for**
9:     Compute $\mathcal{L}_{\text{DML}} = \frac{1}{|\mathcal{B}|} \sum_{b \in \mathcal{B}} \ell(f(\mathbf{x}_{i,b}, \mathbf{x}_{j,k_b}; \theta), y_b)$
10:    Update $\theta \leftarrow \theta - \eta \nabla_\theta \mathcal{L}_{\text{DML}}$
11: **end while**
12: **return** $\theta$

---

*Table 11.* Gap between full- and absent-modality fusion outputs.

|  | KS | AVMNIST | CREMA-D | Food101 |
|---|---|---|---|---|
| $\Delta_{\text{gap}}$ (%) | 3.47 | 1.87 | 3.33 | 5.54 |
| Cos–Sim. | 0.944 | 0.948 | 0.952 | 0.937 |

*Table 12.* Performance Results on UR-Funny datasets.

| Metrics | Visual | Audio | Text | SML | DML |
|---|---|---|---|---|---|
| Accuracy (↑) | 59.42 | 59.28 | 54.15 | 62.57 | **63.78** |
| $\lambda_{\max}$ (↓) | 1.3440 | 1.5302 | 1.4437 | 1.0934 | **1.0920** |
| $\mathcal{L}_{\text{LPF}}$ (↓) | 0.7226 | 0.6823 | 0.7186 | 0.6717 | **0.6705** |

## B.3. Pseudocode for Distributional Multimodal Learning (DML)

In this section, we provide algorithm of DML. Algorithm 1 summarizes the training procedure of DML using the notations introduced in Section 3.1. Given a labeled paired training set $\mathcal{D} = \{(\mathbf{x}_{i,n}, \mathbf{x}_{j,n}, y_n)\}_{n=1}^{N}$, standard multimodal learning uses the fixed pair $(\mathbf{x}_{i,n}, \mathbf{x}_{j,n})$ observed at the same index. DML instead operationalizes the conditional-independence view in Equation (2): for an anchor sample $\mathbf{x}_{i,b}$ with shared semantic label $y_b$, it draws the counterpart modality from the same-label conditional set $\mathcal{I}_j(y_b) = \{k : y_k = y_b\}$, which corresponds to sampling $\mathbf{x}_{j,k_b} \sim p(\mathbf{x}_j \mid y_b)$. The resulting stochastic pair $(\mathbf{x}_{i,b}, \mathbf{x}_{j,k_b}, y_b)$ preserves semantic alignment through $y_b$ while relaxing strict pointwise alignment, and the model $f(\mathbf{x}_i, \mathbf{x}_j; \theta)$ is trained with the same supervised loss $\ell(f(\cdot; \theta), y)$ over these class-consistent pairs.

# C. Additional Experiments

## C.1. Validity of the First-Order Local Approximation

Based on our theories, it is crucial to verify whether its underlying assumptions hold in practice. Specifically, Definition 1 treats the absent-modality input $\bar{\mathbf{x}}_j$ as a reference point and characterizes the transition from $\bar{\mathbf{x}}_j$ to the observed modality input $\mathbf{x}_j$. This formulation relies on the assumption that the induced change in the fusion representation lies within a sufficiently local neighborhood, where the first-order linearization of the fusion function provides an accurate approximation. To validate this, we measure the normalized discrepancy between the observed– and absent–modality fusion outputs as $\Delta_{\text{gap}} = \|\phi(f_i(\mathbf{x}_i), f_j(\mathbf{x}_j)) - \phi(f_i(\mathbf{x}_i), f_j(\bar{\mathbf{x}}_j))\| / \|\phi(f_i(\mathbf{x}_i), f_j(\mathbf{x}_j))\|$. We report the cosine similarity between $\phi(f_i(\mathbf{x}_i), f_j(\mathbf{x}_j))$ and $\phi(f_i(\mathbf{x}_i), f_j(\bar{\mathbf{x}}_j))$ to further examine whether the two fusion representations remain directionally aligned.

As shown in Table 11, the normalized discrepancy remains small across datasets. Moreover, the corresponding representations exhibit high cosine similarity, with most values exceeding 0.9, indicating that the transition from $\bar{\mathbf{x}}_j$ to $\mathbf{x}_j$ does not alter the local geometry of the fusion representation. These results provide empirical evidence that this transition remains within a local regime, thereby supporting the use of Definition 1 as a valid first-order local approximation in practice.

## C.2. Extension to Three Modalities

In this section, we present an extensive evaluation of our method on a 3-modality using the UR-Funny benchmark. This dataset incorporates text, visual, and audio and is specifically designed for humor prediction tasks. Humor is expressed

*Table 13.* **Effect of Data Augmentation (DA) on Multimodal Learning**

| Data Augmentation | | | Kinetics-Sounds | AVMNIST | CREMA-D | UPMC-Food101 |
|---|---|---|---|---|---|---|
| SML | 1 Aug. | Original | 62.95 | 70.35 | 64.95 | 91.48 |
| | | + J | 63.87 | 70.39 | 64.99 | 91.95 |
| | | + C | 63.19 | 70.59 | 64.43 | 92.13 |
| | | + F | 64.14 | 70.87 | 64.68 | 91.99 |
| | 2 Aug. | + J + C | 63.89 | 70.97 | 65.28 | 92.08 |
| | | + J + F | 64.18 | 70.94 | 66.35 | 92.55 |
| | | + C + F | 63.78 | 70.37 | 66.16 | 91.93 |
| | 3 Aug. | + J + C + F | 64.51 | 71.89 | 66.16 | 92.72 |
| DML | 1 Aug. | Original | 65.89 | 71.69 | 65.53 | 92.84 |
| | | + J | 66.49 | 71.22 | 65.70 | 93.57 |
| | | + C | 65.77 | 71.80 | 66.33 | 93.21 |
| | | + F | 66.85 | 72.07 | 66.27 | 94.29 |
| | 2 Aug. | + J + C | 65.68 | 71.95 | 67.45 | 93.72 |
| | | + J + F | 66.15 | 72.47 | 67.60 | 94.49 |
| | | + C + F | 66.64 | 72.14 | 67.32 | 93.86 |
| | 3 Aug. | + J + C + F | **67.19** | **73.01** | **67.75** | **94.93** |

through verbal content (text), gestures and facial expressions (visual), and prosodic cues (audio), with all samples collected from TED Talks. For architectural choices and hyperparameters, we generally follow the settings used for the CREMA-D experiments, with the exception that we utilize a transformer-based model following the design in Hasan et al. (2019).

We provide the experimental results in Table 12, covering classification accuracy, the Hessian maximum eigenvalue ($\lambda_{\max}$) (Ghorbani et al., 2019), and the LPF metric ($\mathcal{L}_{\text{LPF}}$) (Bisla et al., 2022) as described in Section 6.1. As shown in the results, DML improves accuracy by a clear margin (+1.21%) and also achieves flatter minima, as reflected in both the reduced maximum Hessian eigenvalue and lower $\mathcal{L}_{\text{LPF}}$. This provides empirical evidence that the benefits of DML scale naturally to tasks with 3 modalities, consistently outperforming both unimodal and SML.

### C.3. Complementarity of DML and Data Augmentation (DA)

As mentioned on Section 5.3, given that DML introduces stochasticity during training, one might intuitively interpret it as a form of Data Augmentation (DA). Consequently, it is natural to question whether the improvements in accuracy and loss landscape smoothness are merely artifacts of standard augmentation effects rather than a distinct theoretical mechanism. Therefore, in this section, we clarify the relationship between DML and conventional DA. We show that while they share high-level goals, DML and DA is fundamentally different principles, and even complementary approaches.

**Input Space (DA) vs. Output Space (DML)** First, we would like to emphasize the difference between DML and traditional DA. Conventional DA operates on the input space, applying semantics-preserving perturbations (*e.g.*, cropping, rotation, jitter for visual data), where its goal is to expand the distribution within the original semantic boundaries. DML, however, actively constructs such novel pairings; it generates pairs by matching instances that share the same output space.

For example, consider two pairs: `[Poodle Image, Poodle Audio]` and `[Bichon Image, Bichon Audio]`, both labeled as "dog." Traditional DA can only perturb these pairs individually, but cannot generate a novel pair such as `[Poodle Image, Bichon Audio]`. DML, however, can create such a novel pair by leveraging shared labels. This inter-modality stochasticity encourages the model to learn more diverse semantics. Moreover, DML is complementary to conventional DA. One could first generate `[Poodle Image, Bichon Audio]` via DML, and then apply standard DA (*e.g.*, cropping, jittering) to this new pair. This demonstrates that DML introduces a mechanism fundamentally distinct from, yet compatible with, traditional DA.

**Results** To validate the complementarity, we applied DA techniques to the image modality while training both SML and DML. In Table 13, **J** represents Color **J**itter, **C** denotes Cropping, and **F** denotes **F**lipping. For color jitter, we used parameters: brightness = 0.4, contrast = 0.4, saturation = 0.4, hue = 0.1, and a flipping probability of 0.5.

*Table 15.* **Multimodal Adversarial Robustness Evaluation.** Performance is measured by the drop rate (↓) in accuracy compared to the clean accuracy after applying adversarial perturbations to individual modalities. A *lower drop rate* indicates *enhanced robustness*.

*(a)* **Kinetics-Sounds**

|  | Clean Acc. | Modality $i$ Perturbation | Modality $j$ Perturbation |
|---|---|---|---|
| Modality $i$ [A] | 51.95 | 39.73 (23.52% ↓) | – |
| Modality $j$ [I] | 45.70 | – | 8.42 (81.57% ↓) |
| SML | 62.95 | 53.92 (14.34% ↓) | 40.80 (35.19% ↓) |
| DML | 65.89 | **57.56 (12.64% ↓)** | **43.24 (34.38% ↓)** |

*(b)* **AVMNIST**

|  | Clean Acc. | Modality $i$ Perturbation | Modality $j$ Perturbation |
|---|---|---|---|
| Modality $i$ [I] | 64.98 | 15.09 (76.78% ↓) | – |
| Modality $j$ [A] | 41.84 | – | 20.44 (51.15% ↓) |
| SML | 70.30 | 25.81 (63.29% ↓) | 35.85 (49.00% ↓) |
| DML | 71.69 | **29.91 (58.28% ↓)** | **37.79 (34.68% ↓)** |

*(c)* **CREMA-D**

|  | Clean Acc. | Modality $i$ Perturbation | Modality $j$ Perturbation |
|---|---|---|---|
| Modality $i$ [A] | 60.19 | 40.95 (31.97% ↓) | – |
| Modality $j$ [V] | 48.71 | – | 36.11 (25.87% ↓) |
| SML | 64.95 | 46.63 (28.21% ↓) | 54.13 (16.66% ↓) |
| DML | 65.53 | **48.72 (25.65% ↓)** | **55.10 (15.92% ↓)** |

*(d)* **UPMC-Food101**

|  | Clean Acc. | Modality $i$ Perturbation | Modality $j$ Perturbation |
|---|---|---|---|
| Modality $i$ [T] | 86.38 | – | – |
| Modality $j$ [I] | 64.68 | – | 6.86 (89.39% ↓) |
| SML | 91.48 | – | 85.57 (6.46% ↓) |
| DML | 92.84 | – | **87.09 (6.19% ↓)** |

As shown in Table 13, DA generally provides modest improvements for both SML and DML, while DML combined with DA consistently achieves the best performance across all datasets. With all three augmentations (J+C+F), DML improves over its original version by +1.30%, +1.32%, +2.22%, and +2.09% on Kinetics-Sounds, AVMNIST, CREMA-D, and UPMC-Food101, respectively. Furthermore, under the same augmentation setting, DML outperforms SML by +2.68%, +1.12%, +1.59%, and +2.21%, respectively. These results suggest that DML is better able to leverage the increased data diversity introduced by DA, demonstrating its complementary effect with standard augmentation strategies.

More importantly, these results and recent finding support Theorem 2, which suggest that increasing data variance leads to better model performance in multimodal learning: the performance consistently improves as additional DA techniques are incorporated. Notably, DML combined with all three augmentations achieves the highest performance across all benchmarks, confirming the validity of our hypothesis and demonstrating the complementary effect of DA in this context.

### C.4. Adversarial Robustness

In this section, we evaluate the generalization performance of the models. Specifically, we present the results of adversarial perturbation experiments conducted on various multimodal datasets.

**Adversarial Robustness** Adversarial robustness evaluation provides an alternative lens on generalization performance by examining model behavior under worst-case perturbations, where each data point is transformed into its most challenging variant. In this paper, we employ the $\ell_\infty$-Norm Projected Gradient Descent (PGD) adversarial attack—a widely used method for evaluating adversarial robustness, formulated as $\mathbf{x}_{t+1} = \Pi_{\mathcal{B}_\epsilon(\mathbf{x})}\left(\mathbf{x}_t + \alpha \cdot \text{sign}\left(\nabla_\mathbf{x}\mathcal{L}(f(\mathbf{x}_t), y)\right)\right)$ where $\mathbf{x}_t$ denotes the

*Table 14.* PGD Hyperparameter Settings

|  | Image / Visual | Spectrogram⋆ (Audio) |
|---|---|---|
| Step $T$ | 5 | 10 |
| $\epsilon^\dagger$ | 0.1 | 0.005 |
| $\alpha^{\dagger\dagger}$ | 0.001 | 0.001 |

†: Perturbation Range ††: Adversarial Learning Rate ⋆: Spectrogram on KS, AVMNIST & CREMA-D

perturbed input at iteration $t$ (with a total of $T$ iterations), $\mathcal{L}$ is the loss function, $f$ is the model, $y$ is the true label, $\alpha$ is the step size, and $\Pi_{\mathcal{B}_\epsilon(\mathbf{x})}(\cdot)$ is the projection operator onto the $\ell_\infty$-ball of radius $\epsilon$ centered at the original input $\mathbf{x}$. We adopt a modified version of the method from prior work (Yang et al., 2021), with detailed hyperparameter configurations provided in Table 14. For evaluation, we quantify robustness using the drop rate relative to clean accuracy as

$$\textit{Drop Rate} = 100 \times (\text{Clean Accuracy} - \text{Perturbed Accuracy})/\text{Clean Accuracy} \tag{55}$$

**Adversarial Evaluation Results** Tables 15 present the adversarial robustness results across all evaluated multimodal datasets under PGD attacks. Overall, unimodal models show vulnerability to adversarial perturbations. For instance, on AVMNIST (Table 15b), the vision-only model suffers a severe accuracy drop of 76.78%, while the audio-only model drops by 51.15%. Comparable patterns emerge across other datasets: in CREMA-D (Table 15c), the audio-only model drops 31.97% accuracy under perturbation, and in Kinetics-Sounds (Table 15a) the image-only model experiences an 81.57% decrease. These consistent trends highlight the limited robustness of unimodal baselines compared to multimodal models. Moreover, DML outperforms SML under adversarial perturbations, showing smaller accuracy drops for both modalities. This indicates that DML yields enhanced robustness and better generalization compared to standard fusion strategies.

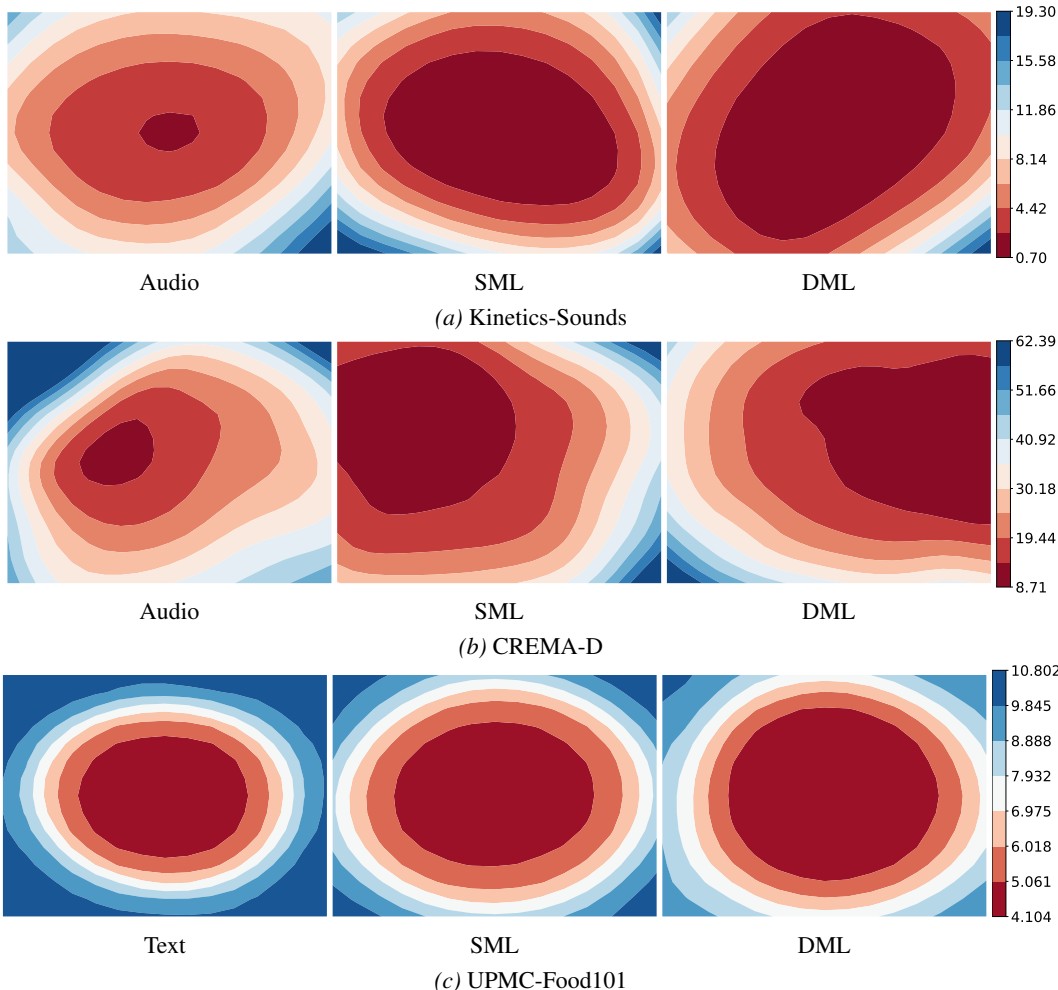

*Figure 5.* Loss landscape visualizations across datasets. Each subfigure corresponds to dataset and compares a unimodal, SML, and DML.

For the UPMC-Food101 dataset, we have not applied adversarial perturbations to the text modality, as PGD attacks on text models are generally considered less natural. Instead, we restricted adversarial evaluations to the image modality. As shown in Table 15d, perturbations applied to the image modality reveal markedly greater robustness under multimodal learning compared with unimodal learning. Although the improvements of DML over SML are modest, DML consistently exhibits smaller performance degradation and higher absolute accuracy. These findings suggest that multimodal learning not only converges to flatter minima but also achieves enhanced robustness relative to unimodal approaches.

### C.5. Loss Landscape Visualizations

In this section, we provide additional loss landscape visualizations via filter normalization (Li et al., 2018) for the datasets used in our experiments, Kinetics-Sounds, CREMA-D and UPMC-Food101, which corroborate our initial hypotheses. Here, Multimodal learning loss landscape (Figure 5), especially DML exhibits a flatter minima compared to the corresponding unimodal learning landscape. Furthermore, DML demonstrates a smoother loss landscape relative to SML, consistent with our expectations. Similar trends are observed across other datasets such in CREMA-D and UPMC-Food101: multimodal learning consistently produces flatter minima compared to unimodal learning. While the difference between DML and SML is marginal, DML still exhibits a slightly flatter landscape. To complement these qualitative observations with quantitative evidence, Table 6 and Figure 4 report generalization metrics of landscape flatness.

### C.6. Revisiting Modality Variance and Gap in Multimodal Learning

**Intrinsic Characteristics of Multimodal Learning** Beyond the comparison between SML and DML, effective multimodal learning depends on the representation geometry of individual modalities. Recent studies on multimodal learning (Lu,

*Table 16.* **Modality variance and Wasserstein Distance (WD) across datasets.** Higher variance implies more diverse features within a modality (intra-modality), while higher WD indicates larger distributional discrepancy between modalities (inter-modality).

| Datasets | Kinetics-Sounds | | AVMNIST | | CREMA-D | | UPMC-Food101 | |
|---|---|---|---|---|---|---|---|---|
| | Audio | Visual | Image | Audio | Audio | Visual | Text | Image |
| **Variance** (ratio) | 0.480 | 0.360 | 1.277 | 0.971 | 0.378 | 0.187 | 1.927 | 0.935 |
| | (1.347) | | (1.316) | | (2.027) | | (2.060) | |
| **WD** | 0.518 | | 0.517 | | 0.543 | | 0.595 | |

2023), as well as Remark 2.2, suggest that both within-modality variance and cross-modality gap play important roles in multimodal fusion. A desirable multimodal representation should preserve sufficiently diverse modality-specific features, while maintaining a balanced gap between modalities. If the gap is too small, modality-specific information may collapse; if the gap is too large, the two modalities may become difficult to align and fuse, leading to degraded performance.

This perspective is consistent with our kernel-based interpretation. In Section 6.3, we analyzed the variance of fused multimodal representations to examine whether DML induces broader feature-space dispersion than SML. However, fused-feature variance does not directly reveal how each individual modality is distributed, nor how the two modality distributions are positioned relative to each other. Therefore, in this section, we provide a more fine-grained dataset-level analysis by separately measuring within-modality variance and cross-modality gap.

Specifically, we consider two complementary quantities. First, *modality variance* measures the dispersion of latent representations within each modality. Although this does not directly compute the variance of the scaled-shift kernel, it serves as an indirect proxy for the effective feature dispersion induced by the kernel. Second, *modality gap* measures the distributional discrepancy between modality-specific representations, which we quantify using the Wasserstein Distance (WD). Together, these quantities allow us to examine whether strong multimodal performance is associated with sufficiently diverse modality representations and a balanced inter-modality gap.

This analysis does not aim to show that larger variance or larger modality gap is always beneficial. Rather, it highlights an important trade-off in multimodal learning: sufficient within-modality variance can provide diverse representations for fusion, but excessive variance imbalance or overly large modality discrepancy may hinder effective cross-modal interaction. Accordingly, Table 16 reports the variance of each modality, the variance ratio between modalities, and WD across datasets.

**Modality Variance and Gap** Before the results, we provide how we measure these variance and gap. Directly comparing the input data across modalities is challenging due to the differing dimensionalities. We therefore analyze the latent vectors from each modality-specific encoder, which are projected into a same dimension of latent space. Although these representations may not fully capture the properties of original data, they could still serve as a reasonable proxy that reflects key characteristics of each modality.

To ensure fair comparisons across modalities, we normalize each latent representation vector $\mathbf{z}_i \in \mathbb{R}^d$ to have unit norm, *i.e.*, $\|\mathbf{z}_i\|_2 = 1$. Given a set of $N$ such unit-norm vectors $\{\mathbf{z}_i\}_{i=1}^N$ from a particular modality, we compute the empirical covariance matrix: $\Sigma = \frac{1}{N} \sum_{n=1}^N (\mathbf{z}_n - \bar{\mathbf{z}})(\mathbf{z}_n - \bar{\mathbf{z}})^\top$, where $\bar{\mathbf{z}} = \frac{1}{N} \sum_{n=1}^N \mathbf{z}_n$. To summarize the variance of each modality, we use the trace $\text{Tr}(\Sigma)$, which captures the average directional spread. To quantify the *modality gap*, we compute the squared 2-Wasserstein distance between the empirical distributions $\mathcal{P}_1$ and $\mathcal{P}_2$ in the shared latent space:

$$W_2^2(\mathcal{P}_1, \mathcal{P}_2) = \inf_{\gamma \in \Gamma(\mathcal{P}_1, \mathcal{P}_2)} \int_{\mathcal{Z} \times \mathcal{Z}} \|\mathbf{z}_1 - \mathbf{z}_2\|_2^2 \, d\gamma(\mathbf{z}_1, \mathbf{z}_2), \tag{56}$$

where $\gamma(\mathcal{P}_1, \mathcal{P}_2)$ is the set of all joint couplings with marginals $\mathcal{P}_1$ and $\mathcal{P}_2$. We selected the WD for its ability to capture the geometric structure of spaces and to provide a quantifiable measure of *how different two distributions are*.

**Relating Modality Geometry to Empirical Performance.** Table 16 shows that Kinetics-Sounds and AVMNIST have relatively balanced modality variances, with variance ratios of 1.347 and 1.316, respectively, and moderate WD values around 0.52. This suggests that both datasets maintain sufficient within-modality diversity without exhibiting excessive cross-modality discrepancy. Such a balanced modality geometry is consistent with the stronger multimodal gains and improved flatness behavior observed in Tables 2 and 6.

In contrast, CREMA-D and UPMC-Food101 exhibit larger variance imbalance, with variance ratios of 2.027 and 2.060, respectively. UPMC-Food101 also shows the largest WD value, 0.595, indicating a larger discrepancy between modality-

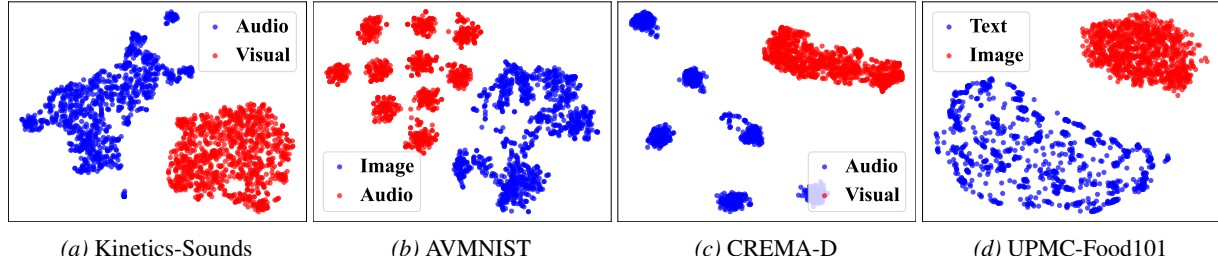

*Figure 6.* **Modality Gap Visualization**: We depict the modality gap via t-SNE visualization. The **blue scatter points** represents dominant modality data that contribute to higher performance, while the **red scatter points** correspond to the modality with lower performance.

specific distributions. Although these datasets still benefit from multimodal learning, the gains are more marginal, suggesting that excessive variance imbalance or modality discrepancy may limit the effectiveness of multimodal fusion. Overall, these observations support our view that multimodal learning benefits not simply from larger variance or larger modality gap, but from a balanced representation geometry with sufficient intra-modality diversity and controlled inter-modality discrepancy.

**t-SNE Visualization** To enhance understanding, we provide t-SNE visualizations in Figure 6, which illustrated the intra- and inter-modality distributions for each dataset. These visualizations aligned the quantitative trends reported in Table 16, showing that datasets with more distinct intra- versus inter-modality characteristics are clearly separated in the latent space. For instance, Kinetics-Sounds and AVMNIST demonstrate relatively well-balanced modality gaps and well-distributed embeddings, which aligns with the results in Table 16. In contrast, CREMA-D and UPMC-Food101 show noticeably uneven dispersion across modalities, with one modality being more scattered than the other, leading to slightly imbalanced.

## C.7. Out-of-Distribution Image-Text Retrieval

In this section, we present further experiments on the Image-Text retrieval task, with a particular focus on Out-of-Distribution (OOD) scenarios. Following the retrieval protocol described in Section 5.4, we fine-tune on Flickr30k and then perform zero-shot evaluation on the unseen MS-COCO dataset (Lin et al., 2014) (Flickr30k → MS-COCO). As expected, On Table 17, DML consistently surpasses SML

*Table 17.* Retrieval on Flickr30k → MS-COCO

| Tasks | Image-to-Text (I2T) | | Text-to-Image (T2I) | |
|---|---|---|---|---|
| **Metrics** | R@1 | R@5 | R@1 | R@5 |
| SML | 56.16 | 78.54 | 36.58 | 64.11 |
| **DML** | **58.84** | **82.72** | **37.83** | **65.38** |

on the unseen MS-COCO distribution (for instance, +2.68% gain in I2T R@1). We regard these findings as a compelling proof-of-concept (PoC) that DML also achieves improved performance in OOD or cross-dataset settings. Here, we used learning rate of $5 \times 10^{-7}$ to finetune.

