# OpenReview forum: "Understanding Multimodal Learning: A Loss Landscape Smoothness Perspective"
_ICML.cc/2026/Conference — ICML 2026 regular_

### Official Review · Reviewer_x6eU · 2026-02-16

**Soundness:** 3
**Presentation:** 3
**Significance:** 3
**Originality:** 3
**Overall Recommendation:** 5
**Confidence:** 4

**Summary:**

This paper proposes a theoretical framework via convolutional smoothing to explain multimodal learning enables a smoother loss landscape and therefore stronger robustness and generalization compared to unimodal learning. Motivated by this theory, the paper proposes Distributional Multimodal Learning (DML), which transforms weak alignment to more enriched supervision by learning from stochastically paired multimodal samples. The theory and the effectiveness of DML are extensively tested on 4 multimodal benchmarks covering common modalities and supported by various analyses.

**Compliance With Llm Reviewing Policy:**

Affirmed.

**Final Justification:**

The authors' rebuttal have addressed all my 2 major concerns in the original review:
1. connect the theory intuition to a stronger notion of generalization (to out-of-domain tasks);
2. provide some explanation of the task-dependent variance in effectiveness of the proposed DML;

Therefore, I raised my rating from 4 to 5 as well as the significance rating to 3, as their additional OOD results demonstrate a broader impact of this theory in motivating new design of multimodal training.

**Key Questions For Authors:**

1. **The major concern that the reviewer has about the submission is regarding the connection between the theoretical framework to the current practices**. Although the authors have made several efforts through extensive experiments and analyses confirming the transfer from the theory to empirical results, the setting remains a little weak as it only concerns in-distribution generalization and robustness to perturbation (which the reviewer agrees, are the direct results of flat loss landscape). The paper could be much more strengthened by having a few more results / discussion about the implications or potential extension of the theoretical framework on the current practices of multimodal learning (e.g. potential connection to a different multimodal pretraining paradigm, as suggested in Originality weaknesses). **The reviewer is willing to raise the score to 5 if this concern is resolved**.
2. Why does DML show different effectiveness when applied to different tasks (e.g. less effective on CREMA-D compared to others, as demonstrated in Table 2, 6, Figure 4)? What is the characterization of limitation where the theory may not effectively transfer to empirical improvements or exhibits variance in practice?

**Limitations:**

The current paper does not suggest a strong connection of how the theoretical framework can be similarly applied to the modern practices of multimodal learning. The paper should make explicit the limitations of the theory framework and have some characterization of how the theory may exhibit different effectiveness when transferred to empirical improvements.

**Strengths And Weaknesses:**

**Soundness**

Strengths:
- There are two major assumptions of the main theorems: 1. the ATI property and 2. conditional independence with respect to an underlying shared semantics (e.g. labels $y\in\mathcal{Y}$). The first assumption is reasonably well-justified with Table 1, while the second assumption seems reasonable under a general multimodal learning setting;
- The math proofs of the main theorems look correct;
- Experiments, ablations, and analyses are relatively extensive to support the claim about loss landscape flatness, improved performance and robustness to perturbation of DML;

Weaknesses:
- Table 2 and 4 have shown that different effectiveness of DML for different tasks / datasets, but it is **unclear whether this should be attributed to the nature of tasks / data satisfies the theoretical assumptions to a different extent or there could be further characterization of where the main theorems apply and how well they transfer to empirical results**;

**Presentation**

Strengths:
- The paper did a good job presenting the theory with high-level interpretations and connect it to empirical observations, so the narrative is easy to follow;
- Experiment details have been carefully documented;

Weaknesses:
- **Figure 2 as the overview of the work can be placed earlier in the pages to help understanding the narrative**;
- It would be very useful to **explicitly list the type of late fusion used in the main experiments**; though it has been made clear from Table 1 that the theoretical framework is compatible with modern architecture like cross-attention, it is not clear from the writing that the experiment results (section 5) support this claim;
- **Figure 4 is not labeled with tasks for each subplot**;
- **Appendix D.2 lists some interesting observations regarding modality variance, modality gap and the loss landscape, but this is not mentioned anywhere in the main text**; it could be useful to have (at least) one sentence summarizing the take-away message, either in the results or the discussion section.

**Significance**

Strengths:
- This paper contributes to building theoretical understanding of the loss landscape of multimodal learning compared to unimodal learning, which is an under-explored direction and provides some fundamental understanding of multimodal learning.

Weaknesses:
- The "in-distribution" statistical generalization and robustness to perturbation seem to be a weaker notion of generalization and the models used in the main experiments are toward smaller scales, making the finding **less connected to modern practices of multimodal learning**. One potential suggestion is:

The reviewer believes **how DML leverages weak alignment and transforms it into additional supervision may contribute to a more stable and generalizable multimodal pretraining paradigm** (which, though is not discussed in this paper, may still has a larger implication in modern practices, and the reviewer strongly encourages the authors to explore deeper in that direction). For example, training on larger-scale pretraining corpus (the reviewer is aware that such suggestions may be unrealistic for a limited compute, but e.g. multi-task training on a mix of datasets could suffice a proof-of-concept) and showing models trained under DML improve generalization or finetuning efficiency on a few held-out relevant downstream tasks compared to SML. (The reviewer is aware that this suggests a stronger notion of "generalization" compared to in-distribution / statistical generalization and requires careful selection of tasks to be aligned with the theory.)

**Originality**
- Understanding multimodal learning via an analysis on loss landscape is a novel perspective, and the proposal of DML is well-motivated by the theory (even though it may not be strictly aligned).

---

> ### Author Rebuttal · Authors · 2026-03-30
>
> > ### **Soundness W. & Concern 2**
>
> For clarity, DML is not introduced as an ad hoc sampling strategy, but as a theory-motivated implication of our smoothing-based view of multimodal learning. Specifically, we first analyze standard multimodal learning (SML) theoretically, and DML follows as a natural extension for further enhancing the smoothing effect predicted by the theory.
>
> At the same time, our theorems do not imply uniform empirical gains across all tasks. When one modality is much less informative (highly imbalanced), the induced kernel becomes less effective, so the benefit of DML can be smaller. Recent studies [1,2] suggest that **CREMA-D exhibits strong modality imbalance**.  This is also consistent with our analysis in Table 2,12: both **CREMA-D and UPMC show imbalanced variance ratios**, but their empirical behavior differs. On CREMA-D, both the **unimodal $\rightarrow$ SML** gain and the **SML $\rightarrow$ DML** gain are marginal, consistent with its **small absolute variance**. By contrast, although UPMC also shows a large variance ratio, its **absolute variance is much larger**, which may explain why the same limitation does not arise in the U.F-101.
>
> To make this point clearer, we analyze the smoothness behavior by **comparing feature variance under SML and DML**, motivated by our theoretical expectation that DML induces larger effective variance:
>
> | | KS | AVMNIST | CREMA-D | U.F-101 |
> | - | - | - | - | - |
> | SML  | 0.161 | 0.055  | 0.0440  | 0.724  |
> | **DML** | **0.200** | **0.079** | **0.0477** | **0.796** |
>
> Since CREMA-D is imbalanced, the variance slightly increased on CREMA-D but marginal, even compared with U.F-101, suggesting that strong modality imbalance limits how much DML can further exploit its smoothing advantage. In the revision, we will clarify this applicability boundary of DML more explicitly.
>
> [1] Wei, Shicai, Chunbo Luo, and Yang Luo. "Improving multimodal learning via imbalanced learning."ICCV’25
>
> [2] Xu, Shaoxuan, et al. "BalanceBenchmark: A survey for multimodal imbalance learning." arXiv 25
>
> &nbsp;
>
> > ### **Presentation W.**
>
> We used a standard concatenation-based late-fusion architecture, which are among the most basic and widely adopted settings in multimodal learning. Our intention was to isolate and study the core effect of multimodal learning, especially the difference between unimodal, SML, and DML, without introducing additional consideration from more complex fusion mechanisms. Yet, following the reviewer’s suggestion, we evaluated a additional fusion module based on **cross-attention**, and we observed the same overall trend:
>
> | | KS | AVMNIST | CREMA-D | U.F-101 |
> | - | - | - | - | - |
> | SML  | 68.85  | 72.76 | 59.95 | 91.39 |
> | **DML**  | **70.32**  | **74.99** | **60.59** | **93.11** |
>
> These additional results show that the benefit of multimodal learning over unimodal learning remains, and that DML consistently improves over SML even with a more modern fusion mechanism such as cross-attention. We will include in the revision and explicitly list the fusion type used in each main experiment for clarity.
>
> &nbsp;
>
> > ### **Significance W. & Major Concern 1 (PoC)**
>
> To address the reviewer’s point that stronger notions of generalization would strengthen the paper more directly, we provide an additional out-of-distribution (OOD) proof-of-concept experiment in a retrieval setting. Specifically, following the retrieval protocol used in our paper, we fine-tuned on Flickr30k and evaluate zero-shot on the unseen MS-COCO dataset **(Train: Flickr30k $\rightarrow$ Eval: MS-COCO)** This directly extends our current Flickr30k retrieval setting toward a stronger cross-dataset transfer (OOD) scenario.
>
> | Flickr30K $\rightarrow$ MS-COCO | I2T | | T2I | |
> | - | - | - | - | - |
> |  | R@1   | R@5   | R@1   | R@5   |
> | SML  | 56.16 | 78.54 | 36.58 | 64.11 |
> | **DML**  | **58.84** | **82.72** | **37.83** | **65.38** |
>
> As hypothesized, DML consistently outperforms SML on the unseen MS-COCO distribution (e.g., +2.68% I2T R@1). We view these results as a meaningful PoC that DML also performs better in OOD or cross-dataset settings. Additionally, we also note that Table 11 in the main paper offers complementary robustness evidence through adversarial evaluation, which is closely related to distribution shift. ***Due to space limitation***, we will include the experimental setup, results, and a broader discussion of these implications in the revision.
>
> As the reviewer kindly noted that they would be open to raising the score if this concern is resolved, we hope this PoC sufficiently addresses the main concern, and we would sincerely appreciate reconsideration of the score.
>
> &nbsp;
>
> > ### **Suggestions (Presentation W.)**
>
> We thank the reviewer for these helpful suggestions. In the revision, we will place Fig.2 earlier to improve readability, add task labels to each subplot in Fig.4, and briefly summarize the main take-away of Appendix D.2 in the main text.

---

> > ### Author Rebuttal · Reviewer_x6eU · 2026-03-31
> >
> > Thanks the author for the comprehensive rebuttal. I'm very satisfied with the additional experiments and evidence to my 3 main concerns addressed above. I raised my overall score to 5 and the significance score to 3 because of the additional OOD results.

---

> > > ### Author Response · Authors · 2026-04-01
> > >
> > > We thank the Reviewer for **thoughtful consideration and for raising the evaluation scores, reflecting that most of the concerns have been resolved.** We are very pleased to hear that our responses have addressed all of their concerns and that the reviewer supports the paper’s acceptance.
> > > We will ensure that all constructive suggestions and discussion points are reflected in the revised manuscript.
> > >
> > > We once again sincerely **thank the reviewer for the insightful discussions and thoughtful consideration**.
> > >
> > > **Best regards, the authors**

---

### Official Review · Reviewer_wvmk · 2026-03-07

**Soundness:** 3
**Presentation:** 3
**Significance:** 3
**Originality:** 3
**Overall Recommendation:** 4
**Confidence:** 2

**Summary:**

This paper studies why multimodal learning often outperforms unimodal learning. The authors explain this from a loss-landscape smoothness perspective and propose a simple training strategy based on this idea. They provide detailed theoretical analysis linking multimodal learning to a smoothing effect over the unimodal objective. Experiments on multiple multimodal datasets show improved performance and smoother optimization behavior.

**Compliance With Llm Reviewing Policy:**

Affirmed.

**Final Justification:**

After considering both the paper and the authors’ rebuttal, I recommend acceptance. The paper is well written, technically interesting, and studies an important question in multimodal learning from a novel optimization perspective. The rebuttal addressed my main concerns reasonably well by adding stronger evidence on the smoothing mechanism, including variance and spectral analyses, and by providing results with a stronger modern backbone. Overall, the rebuttal increased my confidence in the paper’s soundness and significance, and supports a positive final recommendation.

**Key Questions For Authors:**

Please refer to the Weaknesses section above.

**Limitations:**

Please refer to the Weaknesses section above.

**Strengths And Weaknesses:**

# Strengths
1. Well written and easy to follow.
2. The paper focuses on an important and under-explored problem (understanding why multimodal learning works from an optimization perspective) and this angle is novel.
3. The theoretical development is thoughtful and technically interesting, especially the connection between multimodal learning and loss-landscape smoothness.

# Weaknesses
1. The empirical results do support the high-level observation that multimodal training can improve performance and lead to flatter solutions. However, the paper’s key theoretical claim is stronger than that: it argues that multimodal learning works through a convolution-like smoothing mechanism. This part is not directly tested. For instance, the paper does not try to estimate the induced kernel, examine its variance, or show more directly that the multimodal objective behaves like a smoothed version of the unimodal one. Because of this, the experiments mostly confirm the consequence of the theory, rather than the mechanism itself.
2. The paper covers several modality pairs, which is good, but most of the benchmarks and model backbones are still fairly small or conventional. Since the motivation is partly framed around recent progress in large-scale multimodal learning, I would have liked to see at least some evidence on stronger transformer-based multimodal models or more modern large-scale settings. Without that, it is hard to judge how well the main conclusions carry over to today’s multimodal systems.
3. The paper does not provide direct empirical evidence showing that high-frequency components are indeed being suppressed, or that the proposed DML formulation strengthens this effect in a measurable spectral sense. As a result, this part reads more like a plausible explanation than a firmly validated conclusion.

---

> ### Author Rebuttal · Authors · 2026-03-30
>
> > ### **W.1,3: Smoothing Effect of DML**
>
> As suggested, we additionally analyze the smoothness behavior by **measuring feature variance under both SML and DML**, motivated by our theoretical expectation that DML induces larger effective variance. Specifically, we measure the variance of modality across concatenated features $[f_i(x_i), f_j(x_j)]$ to show the large kernel variance of DML's sampling from $p(x_i,x_j|y)$.
>
> | | KS | AVMNIST | CREMA-D | U.F-101 |
> | - | - | - | - | - |
> | SML  | 0.161 | 0.055  | 0.0440  | 0.724  |
> | **DML** | **0.200** | **0.079** | **0.0477** | **0.796** |
>
> As shown in the above table, DML yields higher feature variance than SML across all datasets, consistent with our view that larger variance kernel of DML leads to the larger diversity of features. Then let us move to the smooth effect of DML.
>
> Additionally, in response to the reviewer’s concern regarding frequency analysis, we further provide both existing and new empirical evidence showing that DML strengthens the convolutional smoothing effect. We analyze this from three perspectives:
>
> 1. **Loss Curvature Perspective:** We first re-emphasize the smoothness-related results already reported in **Table 6**. The maximum Hessian eigenvalue $(\lambda_{\max})$ reflects the sharpness of local loss curvature. Our results show that multimodal learning converges to a smoother loss landscape than unimodal learning, and that DML consistently yields a smaller $\lambda_{\max}$ than SML, indicating flatter local curvature.
> 2. **Frequency Domain Perspective 1 (LPF)**: This trend is also supported by the Low-Pass Filter (LPF) analysis in **Table 6**. In this evaluation, a Gaussian filter is applied to the model parameters and we measure the resulting performance variation ($\mathcal{L}\_{\texttt{LPF}}$ and $\Delta\_{\texttt{LPF}}$). Multimodal learning shows stronger robustness under this filtering than unimodal learning, with DML performing best overall. This suggests that multimodal learning is less sensitive to high-frequency perturbations in the loss landscape, and that this property is further strengthened by DML.
> 3. **Frequency Domain Perspective 2 (FFT)**: To directly address the reviewer’s suggestion for a measurable spectral analysis of the smoothing mechanism, we additionally quantified the spectral properties of the loss landscape using a standard signal-processing metric: the **High-Frequency (HF) Power Ratio**. Specifically, we extracted a 1D loss curve by evaluating the model along a layer-wise filter normalized random direction in parameter space [1], and then applied the Fast Fourier Transform (FFT). We computed the HF Power Ratio, defined as the proportion of spectral energy contained in the upper-frequency bands (high-frequency threshold ratio = 0.2). As shown in the table below, DML consistently achieves a lower HF ratio ($\downarrow$) than SML and unimodal baselines:
>
> | Model  | KS | AVMNIST | CREMA-D | UPMC-Food101 |
> | - | - | - | - | - |
> | Modality i | 3.70 | 11.42 | 19.67 | 3.38 |
> | Modality j | 4.64 | 12.22 | 11.71 | 3.70 |
> | SML | 2.89 | 8.01 | 7.84  | 2.10  |
> |**DML**  | **1.98** | **4.47**  | **7.01** | **0.22** |
>
> To recap, we additionally **analyze feature variance in SML and DML** and **perform an FFT-based spectral analysis**, both of which support that DML leads to smoother behavior and more strongly suppresses high-frequency components than SML. We will include these results in the revision.
>
> [1] Li, Hao, et al. "Visualizing the loss landscape of neural nets." NeurIPS'18
>
> &nbsp;
>
> > ### **W.2: Modern Multimodal Learning Model Architecture**
>
> To address the reviewer’s suggestion in a larger multimodal setting, we note that UPMC-Food101 is already included in our paper and is the largest image-text benchmark considered in our experiments, containing 90k image-text pairs across 101 classes. It has also been widely used in recent multimodal classification studies as a representative large-scale image-text benchmark [1,2]. For the model, we further scale the visual encoder from **ResNet-18** to **ViT-B/16** in response to the reviewer’s request for a stronger model, where text model BERT-base is already transformer-based encoder.  Under this configuration, the performance is as follows:
>
> |  | Text  | Image | SML   | DML   |
> | - | - | - | - | - |
> | U.F 101 | 86.38 | 67.96 | 93.01 | **93.78** |
>
> These results indicate that the advantage of DML persists even with stronger modern backbones. Moreover, Table 5 in the main paper demonstrates that DML is also effective for CLIP-based image-text retrieval, further suggesting that our framework extends beyond small conventional benchmarks and remains applicable to modern multimodal settings. We will also add these points in the revision.
>
> [1] Li, Yaowei, et al. "Efficient multimodal fusion via interactive prompting." CVPR'23
>
> [2] Kim, Donggeun, and Taesup Kim. "Missing modality prediction for unpaired multimodal learning via joint embedding of unimodal models." ECCV'24

---

> > ### Author Rebuttal · Reviewer_wvmk · 2026-04-02
> >
> > Thanks the author for the comprehensive rebuttal. I'm very satisfied with the additional experiments and evidence to my 3 main concerns addressed above.

---

> > > ### Author Response · Authors · 2026-04-02
> > >
> > > We sincerely thank the reviewer for the thoughtful evaluation and for acknowledging that the **main concerns, particularly those regarding the smoothing effect of DML and its relevance to modern multimodal learning settings, have been addressed.** We are very encouraged to hear that our responses were helpful and that they satisfactorily resolved the issues raised during the review process. We will carefully incorporate all the reviewer’s feedback and comments into the revised manuscript to further improve its clarity and quality.
> > >
> > > If the reviewer finds that the concerns have been ***adequately resolved***, we would greatly appreciate it if this could be ***reflected in the final overall recommendation and confidence scores.***
> > >
> > > We once again thank the reviewer for the **insightful discussion and valuable suggestions.**
> > >
> > > **Best regards, the authors**

---

### Official Review · Reviewer_DFWg · 2026-03-09

**Soundness:** 3
**Presentation:** 3
**Significance:** 3
**Originality:** 3
**Overall Recommendation:** 4
**Confidence:** 4

**Summary:**

This paper studies why multimodal learning often outperforms unimodal learning from the perspective of loss landscape smoothness. The authors propose a theoretical framework showing that multimodal training can be interpreted as a form of convolutional smoothing of the unimodal loss. Under certain assumptions on the fusion function, the multimodal objective leads to a smoother loss landscape and therefore to flatter minima. Building on this idea, the paper proposes Distributional Multimodal Learning (DML), which uses stochastic pairing of modalities within the same label instead of strict instance-level pairing. Experiments on several multimodal datasets show improvements in accuracy and some empirical evidence of flatter loss landscapes.

**Compliance With Llm Reviewing Policy:**

Affirmed.

**Final Justification:**

Most of my concerns have been addressed during rebuttal phase.

**Key Questions For Authors:**

See Strengths And Weaknesses

**Limitations:**

Yes

**Strengths And Weaknesses:**

The main strength of the paper is that it tries to give a simple and intuitive theoretical explanation for the success of multimodal learning. The convolutional smoothing view is interesting and provides a useful perspective that connects multimodal learning with existing ideas on flat minima and robustness. The paper is also generally well written and the proposed DML training strategy is simple and easy to implement. The empirical section includes several datasets and provides both performance and flatness analysis, which helps support the main claims.

Weakenesses:
1)  The theoretical analysis relies on several assumptions that appear quite strong. In particular, the ATI property may be somewhat unrealistic for modern large-scale architectures. While it may reasonably hold for simple settings such as additive linear fusion or relatively simple non-linear fusion modules, it is less clear whether the property remains valid as the complexity of the fusion mechanism increases. For example, in Section 3.2 the empirical validation on cross-attention blocks is not described in sufficient detail. Important aspects of the experimental setup are missing, such as the number of attention layers used and the number of input tokens for u and v.

2) The theoretical analysis itself is generally sound and rigorous, and the derivations up to Section 4.1 are clear. However, the connection between the theoretical framework and the proposed method introduced in Section 4.2 is not entirely clear. In particular, the link between the convolutional smoothing theory and the DML training strategy appears somewhat indirect. If DML is intended to improve the smoothness of the loss landscape, this connection should be explained more explicitly. Why random pairing within the same class should theoretically enhance smoothness?

3) The idea behind DML appears closely related to the multimodal extension of Supervised Contrastive Learning [1]. In both cases, samples sharing the same label are encouraged to align across different views or modalities.

4) The retrieval experiment raises several concerns from a reproducibility and methodological standpoint. Many implementation details are missing, which makes it difficult to fully evaluate the experiment. In particular, the procedure used to obtain pseudo-labels is not sufficiently justified. The use of UMAP for dimensionality reduction before clustering is unclear. The paper states that correlations and structural relationships are not readily observable in the original high-dimensional space, which seems to be a strong and somewhat counterintuitive assumption. Since clustering methods can in principle operate directly in the original feature space, it is not clear why the additional UMAP step is necessary.

4) Again on retrieval; the choice of K=100 clusters is not well justified. It is unclear why this specific value was selected, especially given the semantic diversity of a dataset such as Flickr30k. The pseudo-labeling procedure could also introduce incorrect semantic alignments. For instance, images depicting visually similar but semantically different situations might be grouped into the same cluster, which could lead DML to align text descriptions with mismatched images. This potential source of noise raises questions about the reliability of the reported retrieval improvements.





[1] Khosla, Prannay, et al. "Supervised contrastive learning." Advances in neural information processing systems 33 (2020): 18661-18673.

---

> ### Author Rebuttal · Authors · 2026-03-30
>
> > ### **W.1: Empirical details**
>
> We apologize for omitting implementation details of the cross-attention block. In Section 3.2, we used a standard multi-head attention module with **1 layer, 8 attention heads**, and **4 input tokens** for each of $u$ and $v$. To address the concern about increasing fusion complexity, we additionally measured the approximation error $\epsilon_{\mathrm{rel}}$ while increasing the depth of the cross-attention module layers:
>
> | attn. layer #  | 1  | 2  | 4|
> | - | - | - | - |
> | $\epsilon_{rel}\times 100 $ (%) | 1.20 | 1.22 | 1.28 |
>
> These results suggest that the local first-order linear approximation remains reasonable even in deeper layers. Nonetheless, we do not assume that ATI holds for all arbitrarily complex fusion models, but for cases that admit a local first-order linear approximation, such cases provided on Section 3.2 and Table 1 in the main paper. We will further clarify the scope of ATI and add results on more complex fusion modules.
>
> &nbsp;
>
> > ### **W.2: Convolutional Smoothing and DML**
>
> In our framework, the paired modality induces an effective smoothing kernel over local shifts of the fused representation, and Remark 4.3.2 indicates that the smoothing strength increases with the variance of this kernel. In SML, the kernel variance is nonzero, but its effective variance is more limited because the pairing is restricted to observed pointwise pairs. DML samples from the broader class-conditional pairing space $p(x_i, x_j \mid y)$ in Eq. (2) through random pairing, thereby increasing the effective kernel variance and strengthening the smoothing effect. This explains **why DML leads to a smoother landscape than SML** under our theoretical framework.
>
> &nbsp;
>
> > ### **W.3: Comparison between DML and SupCon**
>
> DML could be considered related to a multimodal extension of SupCon. However, our main contribution is not this sampling scheme itself, but the *theoretical view that multimodal learning induces smoothing*. Specifically, DML is derived from a theoretical perspective to further amplify that effect by enlarging the label-consistent cross-modal pairing space, resulting in an effective induced kernel variance. This differs from SupCon, which *acts at the objective level by explicitly attracting same-label samples in representation space via loss function*, while *DML acts at the pairing level and is largely loss-agnostic*. Nevertheless, we will clarify this relationship and cite the relevant SupCon literature in the revision.
>
> &nbsp;
>
> > ### **W.4,5: Clarifications on the Retrieval Experiment**
>
> We appreciate the reviewer’s constructive feedback. We want to clarify that $y$ is naturally given by ground-truth class labels. However, in the retrieval setting, such a given $y$ is absent. Therefore, we introduced pseudo-labels for $y$ to examine the effect of DML in the retrieval setting, with the related concerns clarified as follows:
>
> - **Role of UMAP:**  Applying UMAP followed by clustering was not intended as a core theoretical contribution, but only as a practical way to instantiate a proxy $y$ from high-dimensional features. In our setting, the latent feature dimension is 512 and Flickr30k contains roughly 29K samples, which *makes direct clustering computationally expensive, particularly on CPU resources in our environment.* From this perspective, we used UMAP only to obtain a more stable representation for clustering while also reducing computational cost.
> - **Typo on K=100 & Pseudo-Labeling:** We first apologize for a typo in the manuscript: the experiment used **$K=1000$**, not **$K=100$**. We will correct this in the revision. $K=1000$ was selected empirically as a practical hyperparameter to better reflect the semantic diversity of Flickr30k. Additionally, pseudo-labeling can introduce imperfect semantic alignments, including grouping semantically distinct samples together. However, recent multimodal learning results suggest that strictly matching is not always necessary [1], and that learning can still benefit from a certain level of noisy pairings [2,3]. Since our model is based on CLIP-B/16, its representation space is already encouraged to capture task-relevant semantic similarity. In this sense, we regard pseudo-labeling as a reasonable practical approximation of the shared output space $y$.
>
> We acknowledge that we haven't provided enough details about this retrieval experiment. **Due to space limitation**, we will **include all hyperparameters, clustering procedures, and implementation details** for reproducibility in the revision.
>
> [1] Lee, Jae-Jun, and Sung Whan Yoon. "Can One Modality Model Synergize Training of Other Modality Models?." ICLR'25
>
> [2] Huang et al., *MACK: Multimodal Aligned Conceptual Knowledge for Unpaired Image-text Matching*, NeurIPS’22
>
> [3] Byun et al., *MAFA: Managing False Negatives for Vision-Language Pre-training*, CVPR’24.

---

> > ### Author Rebuttal · Reviewer_DFWg · 2026-04-03
> >
> > Thank you to the authors for taking the time to address my concerns. Most of my doubts have been satisfactorily resolved, and I am therefore inclined to increase my score to 4.
> >
> > However, I encourage the authors to carefully revise the experimental section on retrieval, as it is currently not entirely clear and may detract from the overall quality of the paper.
> >
> > Thank you again and good luck.

---

> > > ### Author Response · Authors · 2026-04-03
> > >
> > > We would like to express our sincere appreciation to the Reviewer for the careful evaluation and positive assessment of our work. We are encouraged that **our responses have addressed the Reviewer’s concerns, especially about retreival experiments and smoothing effects of DML**, and contributed to a improved manuscript. We will ensure that the suggestions and comments are thoroughly incorporated into the revised version.
> > >
> > > We are once again **grateful for the Reviewer’s constructive remarks and insightful feedback.**
> > >
> > > **Best regards, the authors**

---

### Official Review · Reviewer_mLny · 2026-03-11

**Soundness:** 2
**Presentation:** 3
**Significance:** 2
**Originality:** 2
**Overall Recommendation:** 4
**Confidence:** 3

**Summary:**

This paper studies multimodal learning from a loss-landscape perspective and argues that multimodal objectives are smoother than unimodal ones. To support this view, the paper provides a convolutional-smoothing interpretation of multimodal loss and introduces a simple training strategy, DML, based on label-conditioned stochastic pairing. Empirically, the method shows consistent improvements over unimodal baselines and standard multimodal learning on several benchmarks.

**Compliance With Llm Reviewing Policy:**

Affirmed.

**Final Justification:**

The response has address my concerns.

**Key Questions For Authors:**

Please refer to the weaknesses above.

**Limitations:**

This work assumes conditional independence between modalities given $y$, which is unlikely to hold for realistic paired multimodal data, but is much closer to the sampling mechanism of the proposed DML setting. As a result, the subsequent theory seems to more directly justify why DML may induce a smoother loss landscape, rather than why multimodal learning in general should be smoother than unimodal learning. This creates a mismatch between the broad motivation of the paper and the actual scope of the theoretical guarantee.

**Strengths And Weaknesses:**

Strengths
1. The paper studies multimodal learning from a loss-landscape viewpoint, which is an interesting and potentially valuable angle.
2. The proposed DML strategy is simple to implement and shows consistent empirical gains across multiple benchmarks.
3. The paper includes not only performance comparisons but also flatness-related analyses, making the empirical study relatively comprehensive.

Weaknesses
1. The use of Approximate Translation-Invariance (ATI) appears conceptually inaccurate. Approximate linearization of the fusion function only implies a local first-order response, not translation invariance in the usual sense. I therefore suggest revising the terminology to avoid overstating what Eq. (1) actually guarantees.
2. Eq. (1) in Section 3.2 is introduced as a **local first-order approximation** under a *small perturbation*. However, the shift in Definition 4.1 compares a real modality input with an absent-modality input, which generally cannot be treated as a small perturbation. Therefore, applying Eq. (1) there seems unjustified. This also makes Theorem 4.2 questionable, since it uses this local approximation to derive a **distribution-level** convolutional result by integrating over the entire $p(x_j\mid y)$, without providing conditions that justify extending a local approximation to a global distributional conclusion.
3. In Section 3.2, $\alpha$ is introduced as a **local Jacobian-related sensitivity** for input perturbations, but in Eq. (5) it is applied directly to an **output difference**. Moreover, if $\alpha$ is indeed a local Jacobian quantity, it should generally depend on the operating point, rather than be treated as a shared term across the whole distribution. This makes the notation and the derivation around Eq. (5) hard to justify.
4. Methodologically, the proposed DML mainly appears to be a simple label-conditioned stochastic pairing strategy. Therefore, the method-level novelty seems limited, and it is unclear whether the main contribution lies in the sampling strategy itself or primarily in the theoretical interpretation built around it.

---

> ### Author Rebuttal · Authors · 2026-03-30
>
> > ### **W.1,2,3: Clarification of ATI and their Local Approximation**
>
> **W.1**: First, we would like to clarify the term `ATI`. Our starting point was not that first-order linearization itself proves translation invariance, but that it provides a tractable starting point for an LTI-inspired interpretation of convolutional smoothing. We didn't expect to claim that ATI holds for every possible fusion model, but for fusion functions that are linear or sufficiently well approximated by a first-order linearization. This is also why we used the term `"approximate"`: the fusion function does not strictly satisfy linearity, but may still exhibit a locally near-linear response.
>
> At the same time, we acknowledge that a genuine `Translation Invariance` claim would require additional consideration beyond linearization. To support this point, we considered several common fusion functions used in recent multimodal learning methods, and Table 1 in the main paper shows that these modules exhibit only small approximation error $\epsilon_{rel}$. We then extended this analysis to a deeper cross-attention fusion module with **8 attention heads** and **4 input tokens** for the further justification, and obtained:
>
> | attn. layer #  | 1 | 2 | 4 |
> | - | - | - | - |
> | $\epsilon_{rel} \times 100$ (%) | 1.20 | 1.22 | 1.28 |
>
> This shows that recent fusion functions are generally well linearized by a first-order linear approximation. However, **we still acknowledge that the our current manuscript is partially overstate this point**. To avoid this ambiguity, we will revise the presentation to clarify that ATI is intended in this approximate sense.
>
> **W.2:** Expanded to the modality case from the previous clarification, we also appreciate that the comparison between a real modality and an absent modality in Definition 4.1 is generally not guaranteed to be a “small perturbation” in the strict theoretical sense. Since it is difficult to characterize this gap theoretically, we additionally provide empirical results on the gap between the real-modality and absent-modality cases as $\Delta_{\texttt{gap}} = || \phi(f_i(x_i), f_j (x_j) - \phi(f_i (x_i),f_j(\bar{x}_j)) ||  / || \phi(f_i(x_i), f_j (x_j)) || $), and cosine similarities between $\phi(f_i(x_i), f_j (x_j))$ and $ \phi(f_i (x_i),f_j(\bar{x}_j))$:
>
> |  | KS | AVMNIST | CREMA-D | Food101 |
> | - | - | - | - | - |
> | $\Delta_{\texttt{gap}} \times 100 $ (%) | 3.47  | 1.87    | 3.33    | 5.23    |
> | Cosine Similarity  | 0.944 | 0.948   | 0.952   | 0.894   |
>
> These results indicate that the gap remains marginal and that the representations remain highly similar in practice, which reasonably supports the Def.4.1. We will add these results and clarify the empirical justification of this gap in the revision.
>
> **W.3**: We make the notation around $\alpha$ more precise. In the revision, we will revise Eq.(5) to write $\alpha=\alpha(x_i,x_j)$, making clear that it denotes a local Jacobian-related sensitivity depending on the operating point, rather than a shared constant over the full distribution. We will also revise the corresponding derivation so that $\alpha$ is used consistently in this local sense.
>
> Additionally, for Theorem 4.2 and Eq.(5), we had implicitly worked under standard regularity and remainder-control conditions, under which, in general, **local first-order approximations can be extended to approximate distribution-level through averaging (local $\rightarrow$ distribution-level average) [1,2]**. However, these conditions were not stated explicitly, but were implicitly assumed. Thus, we will state them explicitly in the revision.
>
> [1] van der Vaart, A. W. "Asymptotic Statistics" Cambridge Univ. Press'98.
>
> [2] Kasy, Maximilian. "Uniformity and the delta method." Journal of Econometric Methods'19
>
> &nbsp;
>
> > ### **W.4 and Limitation: Core Contribution**
>
> Our primary contribution is not merely a sampling strategy, but a theoretical analysis addressing a broader question: ***Why does multimodal learning (SML) generally outperform unimodal learning?*** Our theory first explains why SML yields a smoother loss landscape than unimodal learning, and then extended to DML.
>
> As discussed in **Section 4.2** and **Remark 4.3.2**, the smoothing strength is governed by the variance of this induced kernel. In the unimodal setting, the kernel has essentially zero variance, yielding the weakest smoothing. In SML, this variance becomes nonzero due to cross-modal diversity, which explains why multimodal learning is smoother than unimodal learning. DML then arises as a direct methodological implication of this theory: if stronger smoothing comes from larger effective kernel variance, then expanding to the broader conditional space naturally strengthens the effect. In this sense, our theory first explains **`why SML > unimodal`**, and then extends to explain **`why DML > SML`**. Thus, DML is not merely a simple heuristic, but a method directly motivated by the theory.

---

> > ### Author Rebuttal · Reviewer_mLny · 2026-04-03
> >
> > Thanks for the detailed response, and I have adjusted my score accordingly.

---

> > > ### Author Response · Authors · 2026-04-03
> > >
> > > We sincerely appreciate the Reviewer’s careful consideration and the resulting positive evaluation of our work.
> > > We are very encouraged that **our responses have addressed the Reviewer’s concerns regarding the the validity of our theoretical assumptions** and have contributed to a substantially improved version of the paper. We will ensure that the suggestions and discussion points raised by the Reviewer are fully reflected in the revised manuscript.
> > >
> > > We once again **thank the Reviewer for the constructive discussions and insightful feedback.**
> > >
> > > **Best regards, the authors**

---

### Decision · Program_Chairs · 2026-04-30

**Decision:**

Accept (regular)

**Comment:**

This paper provides a novel theoretical framework grounding the superiority of multimodal learning in convolutional smoothing, which results in a smoother loss landscape and flatter minima. Reviewers initially raised concerns regarding the validity of the Approximate Translation-Invariance (ATI) property in complex architectures, the specific mechanism by which the proposed Distributional Multimodal Learning (DML) enhances smoothness, and the limited scale of initial experiments. I recommend acceptance because the authors provided a comprehensive rebuttal that included empirical validation of ATI on deep cross-attention layers, spectral analysis confirming high-frequency suppression, and new out-of-distribution results on modern backbones. For the final version, the authors should ensure all implementation details for the retrieval experiments and clarified boundary conditions for modality imbalance are explicitly documented.